# Targeting myelin lipid metabolism as a potential therapeutic strategy in a model of CMT1A neuropathy

R. Fledrich [1,2,3], T. Abdelaal [1,4,5], L. Rasch[1,4], V. Bansal[6], V. Schütza[1,3], B. Brügger[7], C. Lüchtenborg[7], T. Prukop[1,4,8], J. Stenzel[1,4], R.U. Rahman[6], D. Hermes [1,4], D. Ewers [1,4], W. Möbius [1,9], T. Ruhwedel[1], I. Katona [10], J. Weis[10], D. Klein[11], R. Martini[11], W. Brück[12], W.C. Müller[3], S. Bonn [6,13], I. Bechmann[2], K.A. Nave[1], R.M. Stassart [1,3,12] & M.W. Sereda[1,4]

In patients with Charcot–Marie–Tooth disease 1A (CMT1A), peripheral nerves display aberrant myelination during postnatal development, followed by slowly progressive demyelination and axonal loss during adult life. Here, we show that myelinating Schwann cells in a rat model of CMT1A exhibit a developmental defect that includes reduced transcription of genes required for myelin lipid biosynthesis. Consequently, lipid incorporation into myelin is reduced, leading to an overall distorted stoichiometry of myelin proteins and lipids with ultrastructural changes of the myelin sheath. Substitution of phosphatidylcholine and phosphatidylethanolamine in the diet is sufficient to overcome the myelination deficit of affected Schwann cells in vivo. This treatment rescues the number of myelinated axons in the peripheral nerves of the CMT rats and leads to a marked amelioration of neuropathic symptoms. We propose that lipid supplementation is an easily translatable potential therapeutic approach in CMT1A and possibly other dysmyelinating neuropathies.

[1] Department of Neurogenetics, Max-Planck-Institute of Experimental Medicine, Göttingen 37075, Germany. [2] Institute of Anatomy, University of Leipzig, Leipzig 04103, Germany. [3] Department of Neuropathology, University Hospital Leipzig, Leipzig 04103, Germany. [4] Department of Clinical Neurophysiology, University Medical Center Göttingen, Göttingen 37075, Germany. [5] Chemistry of Natural and Microbial Products Department, Pharmaceutical and Drug Industries Division, National Research Centre, Giza 12622, Egypt. [6] Center for Molecular Neurobiology, Institute of Medical Systems Biology, University Medical Center Hamburg-Eppendorf, Hamburg 20251, Germany. [7] Heidelberg University Biochemistry Center (BZH), Heidelberg 69120, Germany. [8] Institute of Clinical Pharmacology, University Medical Center Göttingen, Göttingen 37075, Germany. [9] Center for Nanoscale Microscopy and Molecular Physiology of the Brain (CNMPB), Göttingen 37075, Germany. [10] Institute of Neuropathology, University Hospital Aachen, Aachen 52074, Germany. [11] Department of Neurology, Section of Developmental Neurobiology, University Hospital Wuerzburg, Wuerzburg 97080, Germany. [12] Institute of Neuropathology, University Medical Center Göttingen, Göttingen 37075, Germany. [13] German Center for Neurodegenerative Diseases, Tübingen 72076, Germany. These authors contributed equally: R. M. Stassart, M. W. Sereda. Correspondence and requests for materials should be addressed to R.F. (email: fledrich@em.mpg.de) or to K.A.N. (email: nave@em.mpg.de) or to R.M.S. (email: stassart@em.mpg.de) or to M.W.S. (email: sereda@em.mpg.de)

Myelination of axons is essential for rapid impulse propagation and is made by Schwann cells in the peripheral nervous system[1]. Myelin or Schwann cell defects underlie a group of common neurological disorders referred to as demyelinating polyneuropathies (PNP)[2]. Charcot–Marie–Tooth disease 1A (CMT1A) is the most frequently inherited demyelinating PNP caused by the duplication of the gene encoding the peripheral myelin protein of 22kDA (PMP22), an integral transmembrane protein of the myelin sheath[3,4]. The disease is characterized by a slowly progressive nature, and affected patients suffer from a distally pronounced muscle weakness as well as from sensory symptoms. Most patients seek medical advice in the second decade of life[5], but first symptoms such as a mild walking disability are already present in childhood[6]. In line with the early onset, nerve biopsies from children with CMT1A disease[7] as well as the analysis of a genuine CMT1A animal model demonstrate a pronounced dysmyelination early postnatally[8]. We previously used a CMT1A rat model, which closely mimics the human disease[9], to examine peripheral nerve development and could show that Pmp22 transgenic Schwann cells show a strong delay in myelination, with many fibers remaining unmyelinated until adulthood[8]. Consequently, the pool of functional, myelinated axons in CMT1A is markedly reduced throughout life. These pathologically unmyelinated fibers degenerate with time, suggesting that improving myelination would also rescue axon survival[8]. However, it has been mysterious how altered Schwann cell differentiation links mechanistically to reduced myelination in CMT1A[8,10,11].

Schwann cell lipid metabolism was a plausible missing link in CMT1A pathogenesis, as myelin lipids are important for both, myelin membrane growth and long-term integrity. Lipids account for about 70% of the myelin membrane, with phospholipids, cholesterol, and glycosphingolipids being most abundant, comprising 50.6%, 27.2%, and 17%, respectively, of the total lipids in the purified myelin[12]. Importantly, the de novo synthesis of cholesterol by Schwann cells is rate limiting for myelin biogenesis, as demonstrated by mouse mutants with disturbed cholesterol biosynthesis[13,14]. In contrast, mouse mutants in which other (non-cholesterol) lipid-generating enzymes have been ablated show normal myelination, often followed by impaired myelin maintenance[15]. In line, interference with fatty acid synthesis causes more subtle defects in the peripheral nerves, including changes of the myelin ultrastructure as well as an altered Schwann cell metabolism[16]. Indeed, a role for glial lipids in intermediate metabolism, independent of myelin biogenesis, has been suggested[16–18]. Moreover, all major myelin lipids are present in lipid rafts with structural myelin proteins, including PMP22[13,19–21].

We previously observed a transcriptional downregulation of the lipid-related genes in sciatic nerves of Pmp22 transgenic rats[22]. In detail, in a comparative transcriptomic analysis using microarrays between mildly and severely affected CMT rats at early (P6) and late (P90) time points, we found the lipid-associated genes to be differentially expressed between mildly and severely affected CMT rats, which allowed us to derive surrogate biomarkers for disease severity[22,64].

This led to the hypothesis that Pmp22 overexpression in CMT1A disease and disturbed intracellular lipid metabolism interfere with myelin biogenesis and cause the dysmyelinating phenotype. Myelinating glial cells normally self-generate their lipids during development, but they can take up and utilize extracellular lipids[13,23,24]. Moreover, when mice were fed with a special lipid diet, the myelin–lipid composition transformed substantially[25,26]. Thus, Schwann cells respond to internal and external changes of lipid metabolism, rendering lipid supplementation an attractive therapeutic option in diseases such as CMT1A.

## Results

### Schwann cells in CMT1A display impaired lipid biosynthesis.
We previously observed a differential expression of lipid-related genes between mildly and severely affected CMT1A rats at single early (P6) and late (P90) time points using microarray analysis[22]; however, the temporal regulation of lipid metabolism and its relation to postnatal myelination remained unclear. In a first step, we therefore used RNA-sequencing to analyze the lipid biosynthetic and metabolic processes in a longitudinal manner in sciatic nerve transcriptoms derived from wild-type and Pmp22 transgenic rats before and during the time course of myelination (Fig. 1a). Importantly, in this unbiased approach, lipid biosynthetic and metabolic processes were the most prominently downregulated transcripts in CMT1A rats during postnatal development, with a reduced mRNA expression of both lipid catabolizing and anabolizing transcripts (Fig. 1a). The observed transcriptional dysregulation in CMT1A strongly correlates with the time course of myelin biosynthesis, and no major changes of gene transcription were observed at embryonic day 21 (Fig. 1a). In order to unravel the nature of mRNA dysregulation in Pmp22 transgenic Schwann cells, we grouped the individual transcripts into four distinct patterns based on the expression profile similarity (Fig. 1a). In fact, almost all lipid-associated transcripts follow a similar expression pattern with lipid genes failing to be upregulated in Pmp22 transgenic Schwann cells during myelination (Pattern 1, to a lesser extent Pattern 3, Fig. 1a), indicating that Schwann cells display an impaired ability to mount a lipid biosynthetic transcriptional program in CMT1A disease.

Does this explain the compromised ability of CMT Schwann cells to myelinate? We sought to test this in primary Schwann cell dorsal root ganglia (DRG) neuron co-cultures derived from Pmp22 transgenic animals by exogenous lipid supplementation. Indeed, when compared with wild type, the Pmp22 transgenic Schwann cell DRG co-cultures appeared to recapitulate the pronounced dysmyelination observed in vivo in CMT1A disease (Fig. 1b, c). In order to decide for the optimal lipid species in a supplementation experiment, we performed in-depth bioinformatics on the metabolic pathways of major lipid classes in myelin and their perturbation by CMT1A. The vast majority of transcripts for proteins involved in phospholipid, cholesterol, and glycosphingolipid biosynthesis were downregulated (Supplementary Fig. 1A–C). Among the phospholipids, the metabolism of phosphatidylcholine, a major myelin compound, was severely impaired at the transcriptional level in sciatic nerves of CMT1A rats (Fig. 1d).

Importantly, while phosphatidylcholine treatment did not improve myelination when added to lipid-oversaturated standard medium (Supplementary Fig. 1D), the supplementation of Pmp22 transgenic Schwann cell co-cultures with 2 µg/ml phosphatidylcholine (PC) in delipidated medium resulted in a striking increase in the number of myelinated segments, stained for myelin basic protein (MBP) (Fig. 1e and Supplementary Fig. 1E). Notably, phosphatidylcholine treatment also raised the number of myelinated segments in wild-type cultures, ultimately leading to an equal degree of in vitro myelination in control and CMT1A-derived Schwann cell DRG co-cultures (Fig. 1e). In a subsequent experiment, we treated Pmp22 transgenic Schwann cell neuron co-cultures with tail group-labeled phosphatidylcholine (BODIPY-PC). Importantly, Schwann cells incorporated the labeled phosphatidylcholine into the myelin membrane, suggesting that the rescue of myelination is due to a direct incorporation of the exogenously supplied lipids into the myelin sheath (Fig. 1f). Of note, BODIPY alone (tagged to pentanoic acid) was taken up by the cells, but was not incorporated into the myelin sheath (Fig. 1f).

**Phospholipid therapy ameliorates neuropathy in CMT1A rats**. Following this in vitro findings, we tested whether systemically applied phospholipids can reach the peripheral nervous system and are utilized by Schwann cells for myelin biosynthesis in vivo. We injected fluorescently labeled phosphatidylcholine (BODIPY-PC; 500 μg) once into the tail vein of CMT rats during active

myelination (postnatal day 15) and analyzed the myelin sheaths histologically in sciatic nerves 1 week post injection (Fig. 2a). Notably, we were able to detect several myelinated fibers with incorporated BODIPY-PC, confirming that Schwann cells make use of exogenously supplied lipids for myelin biosynthesis (Fig. 2a).

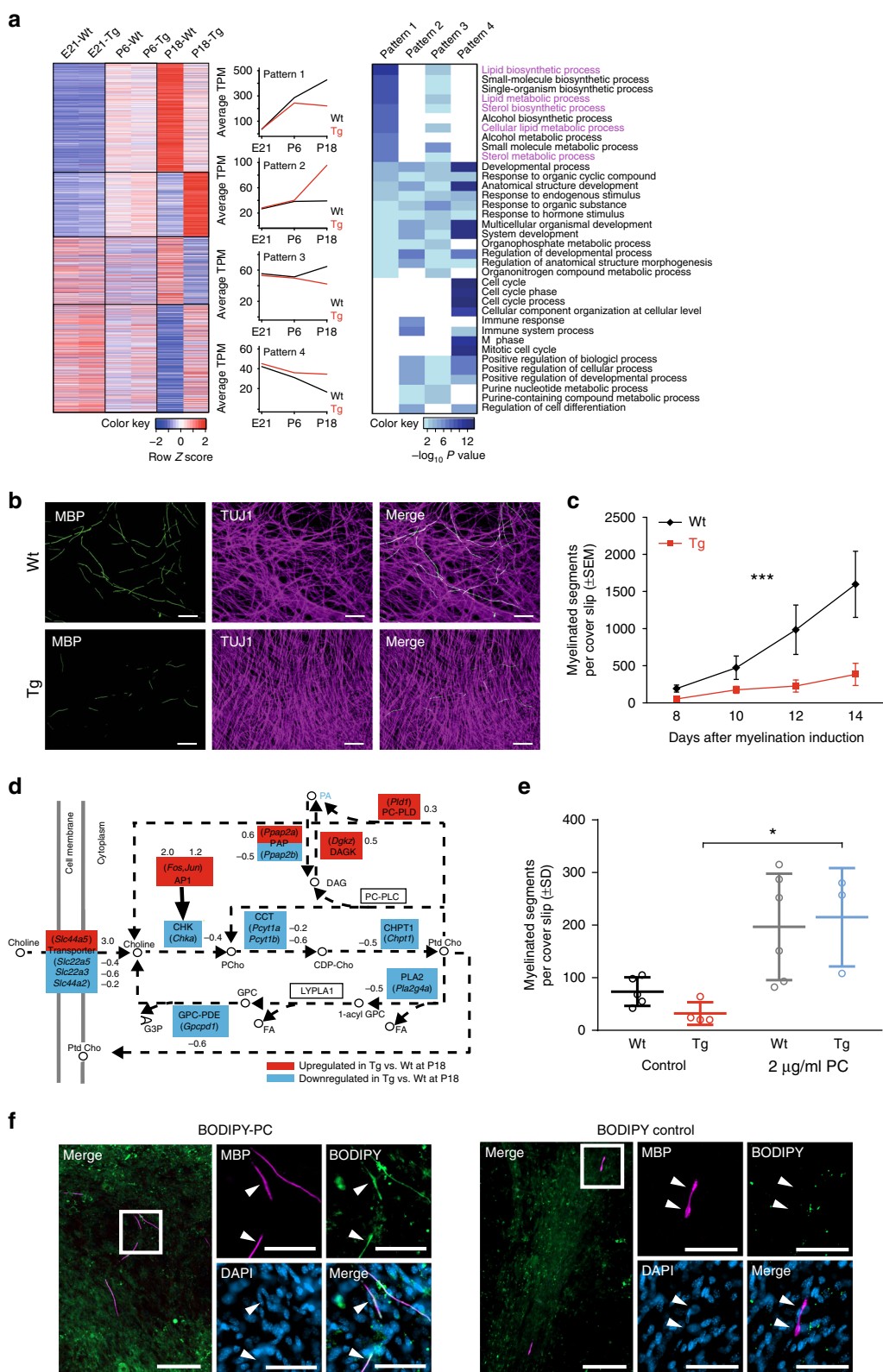

Based on the positive effect of PC treatment on myelination in vitro, we next asked whether lipid supplementation is also able to promote myelination in an experimental treatment approach in the CMT1A rat model in vivo. In a proof-of-principle trial, we treated *Pmp22* transgenic and control rats from postnatal day 2 (P2) until adulthood (P112) by providing either normal food or a diet enriched with two different concentrations of phospholipids (PL) into the cages housing the mother and the pups (Fig. 2b). The experimental diets contained 3.8% crude fat (see Methods) and were enriched by either 0.3 or 3% soy bean-derived PL composed of 55% phosphatidylcholine and 20% phosphatidylethalonamine in addition to trace amounts of other lipids. Of note, the lipid diet was well tolerated by the animals and did not show side effects or abnormal weight gain (Supplementary Fig. 2A).

As rat pups only consume solid food from postnatal day 15 on, we analyzed to which extent the supplemented lipids passed in the milk of the nursing dams. To this end, we treated the nursing dams after they gave birth from P2 to P9 with 3% PL. At P9, we collected the milk from the stomachs of the pups and subjected the samples to mass spectrometry. Of note, the lipid proportion of milk in general consists of >98% of neutral lipids, mostly TAG[27]. With over 60%, the major fatty acid compound in the supplemented phospholipid is linoleic acid, which harbors two double bonds (C18:2, see Methods). When plotting the analyzed milk neutral lipids (TAG plus DAG) as a function of the double bonds, a significant shift toward species with either two or four double bonds was visible, indicating that the supplemented phospholipid fatty acid tails have passed in the mother's milk in the form of neutral lipids (Supplementary Fig. 2B).

Importantly, when we analyzed the grip strength of the animals treated from P2 to P112, we detected a dose-dependent improvement in *Pmp22* transgenic rats treated with both, a 0.3% or a 3% PL diet (Fig. 2c). CMT1A rats that received the 3% PL treatment showed a significantly improved muscle strength at the end point of the therapeutic trial (P112), even reaching wild-type levels (Fig. 2c). In good agreement, the muscle circumference, a measure for muscle mass, showed a significant increase in both, the 0.3% and 3% PL-treated CMT1A rats when compared with controls (Fig. 2d). Furthermore, in electrophysiological recordings at study end, we detected a mild but significant improvement of the nerve conduction velocity (NCV) in addition to a substantial amelioration of the decreased compound muscle action potentials (CMAPs), indicating a larger number of fast propagating myelinated axons after PL therapy[28,29] (Fig. 2e, f). These results prompted us to further investigate the myelination status of peripheral nerves by histology. In the tibial nerve of

PL-treated CMT1A rats, we found a striking increase of myelinating axons, again with numbers reaching wild-type levels in the 3% PL therapy group (Fig. 3a, b). Moreover, when we quantified the density of axonal neurofilaments, which is aberrantly increased in CMT1A rats and has been previously shown to impair nerve function in peripheral neuropathy[30], we found the neurofilament spacing to be improved after PL therapy without changes in axonal diameter (Fig. 3c, d and Supplementary Fig. 3A). Hence, exogenous lipid supplementation emerges as an effective therapeutic approach in a CMT1A rat model.

**Dietary lipids normalize myelin ultrastructure in CMT rats.** Lipid supplementation did not affect the number of unmyelinated axons or onion bulb formations in CMT1A rats (Supplementary Fig. 3B, C), suggesting that PL treatment predominantly supports myelinating *Pmp22* transgenic Schwann cells. Analysis of myelin sheath thickness in untreated CMT rats, as assessed by *g*-ratio quantification, showed the expected distribution of small caliber hypermyelinated fibers and large caliber axons with reduced myelin sheath thickness, however, with no alteration after PL therapy (Fig. 4a, b). Likewise, we observed no improvement of the reduced internodal length in teased fiber preparations of *Pmp22* tg nerves after PL therapy (Supplementary Fig. 3D). In line, when we determined the nodal width with the help of immunohistochemistry against the nodal $Na_V1.6$ and the confining myelin MBP, we found no alteration of an abnormally widened nodal area in treated CMT1A animals (Supplementary Fig. 3E).

When we analyzed the myelin ultrastructure in human CMT1A biopsy material at the electron microscopic level, myelin morphology was overall comparable between CMT1A nerve samples and respective controls (Fig. 4c, d). Of note, non-standardized processing of human nerve biopsies in clinical routine hampers reliable quantification of subtle ultrastructural differences; nonetheless, a mildly widened interperiodic distance may be present in human CMT1A, as quantified in two control versus three CMT1A specimens (Fig. 4d), in line with previous observations[31]. These findings prompted us to assess myelin periodicity in the CMT rat model, where we detected a significant widening of the distance between the individual myelin layers in CMT rat myelin (Fig. 4e, f), in line with the human observations. Importantly, we found the widened myelin periodicity that we observe in CMT rats to be normalized after PL therapy (Fig. 4e, f). In order to assess to which extent the altered ultrastructural myelin morphology may be due to alterations in the protein and lipid composition of the myelin sheath, we performed western blot and mass spectrometric analyses of purified myelin,

---

**Fig. 1** CMT rat Schwann cells display impaired endogenous ability to synthesize lipids. **a** Heatmap (left) shows scaled TPM values (transcripts per million) for differentially expressed genes (adjusted *p* value <0.05 and log₂ fold change >|0.5|) at P18. Genes downregulated in Tg (*Pmp22* transgenic) versus Wt (wild type) at P18 were further divided into two patterns—genes with increasing TPM in Wt from E21 to P18 (pattern 1) and remaining (pattern 3). Genes upregulated in Tg versus Wt at P18 were further divided into two patterns—genes with increasing TPM in Tg from E21 to P18 (pattern 2) and remaining (pattern 4). Heatmap (right) shows top ten biological process terms (from WebGestalt) for each pattern. **b** Dorsal root ganglia neuron and Schwann cell co-cultures from wild-type (Wt) and *Pmp22* transgenic (Tg) mice revealed impaired myelination in Tg co-cultures. Shown are representative pictures of Wt and Tg cultures 14 days after myelination induction. Myelin is immunostained for MBP (green) and axons for TUJ1 (magenta). Scale = 100 μm. **c** Quantification of (**b**) in a timeline from 8 to 14 days after myelination induction. (*n* = 3–8 per group and time point, mean ± standard error of mean (SEM), two-way ANOVA, ***p* value <0.001). **d** Differentially expressed genes involved in choline metabolism in Tg versus Wt sciatic nerves at age P18 (KEGG pathway: choline metabolism in cancer). Up- and downregulated genes (adjusted *p* value <0.05) are shown in red and blue, respectively. Circles represent lipid products. **e** Wt and Tg co-cultures maintain a reduced myelination competence when grown with delipidated serum (compare to (**c**)). Addition of 2 μg/ml phosphatidylcholine (PC) to the culture medium increases myelination in both groups 10 days after myelination induction (*n* = 3–6 per group, mean ± standard deviation (SD), one-way ANOVA, and Sidak's multiple comparison post test, **p* value <0.05). **f** When supplied to the myelination medium, Tg co-cultures integrate BODPIPY-labeled phosphatidylcholine (left panels, BODIPY-PC, green, 2 μg/ml medium) into the myelin sheaths (MBP, magenta), whereas control BODIPY (tagged to pentanoic acid) was not incorporated into the myelin membranes (right panels, 10 days after myelination induction). Cultures were counterstained for nuclei (DAPI, blue). Scale = 100 μm (blow up 50 μm)

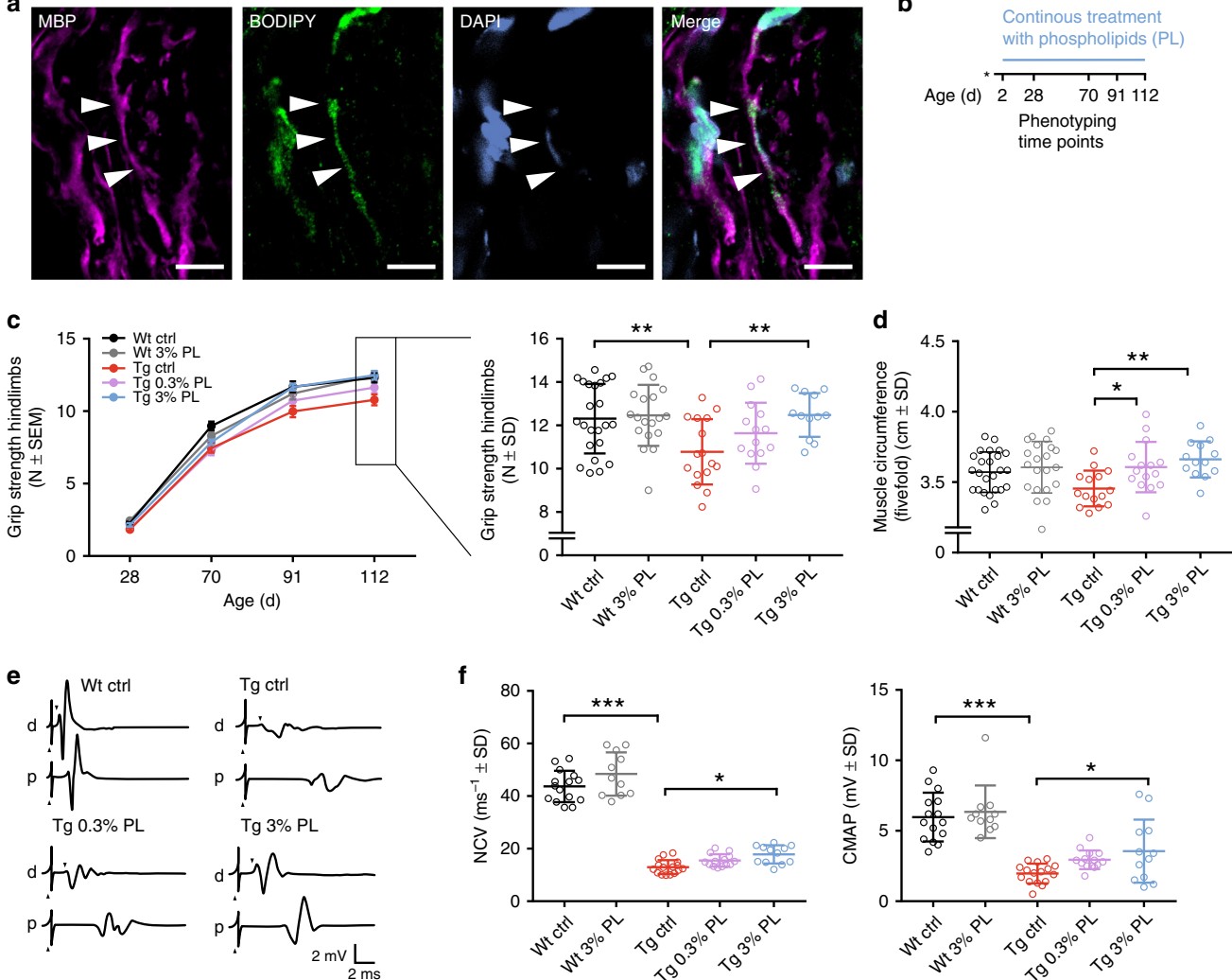

**Fig. 2** Phospholipid therapy in CMT rats ameliorates neuropathy. **a** BODIPY-phosphatidylcholine (BODIPY-PC, green) was injected into the tail vain of a 15-day-old CMT rat. Shown is a longitudinal confocal microscopic image of a sciatic nerve cryo section 7 days post injection. Myelinated fibers (MBP, magenta) display myelin sheaths with incorporated BODIPY-PC (arrow heads, scale = 10 μm). **b** Treatment scheme. CMT rats (Tg) and wild-type rats (Wt) were fed with either normal diet or a food enriched with 0.3 or 3% phospholipids (PL, see Methods section for details) from postnatal day (P) 2 to P112. **c** Grip strength measurements at the age of 28, 70, 91, and 112 days (left panel). For the 112 days time point, the individual data points are shown (right panel). Wt control rats (Wt ctrl, black, n = 22) and Wt rats fed with 3% PL (Wt 3% PL, gray, n = 19) displayed normal motor performance. Tg control rats (Tg ctrl, red, n = 16) are weaker compared with Wt. Tg rats treated with 0.3% PL (Tg 0.3% PL, purple, n = 15) and 3% PL (Tg 3% PL, blue, n = 13) display a stepwise improvement in grip strength (one-way ANOVA). **d** The circumference of the muscles of the lower forelimbs of CMT rats is increased by phospholipid therapy (Wt ctrl, black, n = 25; Wt 3% PL, gray, n = 21; Tg ctrl, red, n = 15; Tg 0.3% PL, purple, n = 15; Tg 3% PL, blue, n = 13, one-way ANOVA, Sidak's post test). **e** Representative traces of electrophysiological recordings at the tail motor nerve at P112. Wild-type rats display a shorter distal motor latency (DML, arrow head down, stimulation artifact is demarcated by arrow head up) and bigger compound muscle action potential (CMAP) amplitudes after distal (d) and proximal (p) stimulation when compared with transgenic control rats (Tg ctrl). Treatment with PL (0.3 and 3%) increases CMAP. **f** Quantification of (**e**). Tg control rats (Tg ctrl, red, n = 16) display reduced nerve conduction velocities (NCV, left panel) and CMAP amplitudes (right panel) when compared with Wt control rats (Wt ctrl, black, n = 15) and Wt rats treated with 3% PL (Wt 3% PL, gray, n = 11). Phospholipid treatment in Tg rats (Tg 0.3% PL, purple, n = 14; Tg 3% PL, blue, n = 12) increases NCV and CMAP (one-way ANOVA, Sidak's post test, p value *<0.05 and **<0.01 and ***<0.001, standard deviation (SD), standard error of mean (SEM))

comparing wild-type rats to PL-treated and non-treated CMT rats (Fig. 5a–d). Here, we did not detect major differences in protein stoichiometry between PMP22 and MPZ from untreated and PL-treated CMT rats, and when compared with wild-type controls (Fig. 5a, b). Interestingly, MBP protein levels in myelin were slightly increased after PL therapy in CMT rats, but remained unaltered in full nerve lysates (Fig. 5a, b and Supplementary Fig. 4A).

However, when normalized to protein input, mass spectrometric analysis of purified myelin demonstrated a highly significant reduction by >50% of nearly all classes of lipids in CMT rat myelin (Fig. 5c). Importantly, after PL therapy, the lipid-to-protein ratio in the myelin of CMT rats showed a strong trend toward normalization, although the majority of individual lipid classes does not reach significance compared with non-treated *Pmp22* transgenic controls (Fig. 5c). Indeed, the observed high variability in adult *Pmp22* transgenic rats at onset may underlie the lack of significance after treatment and may reflect the ongoing de- and remyelination in adult peripheral nerves of CMT1A rats. Interestingly, the stoichiometry of the main lipid

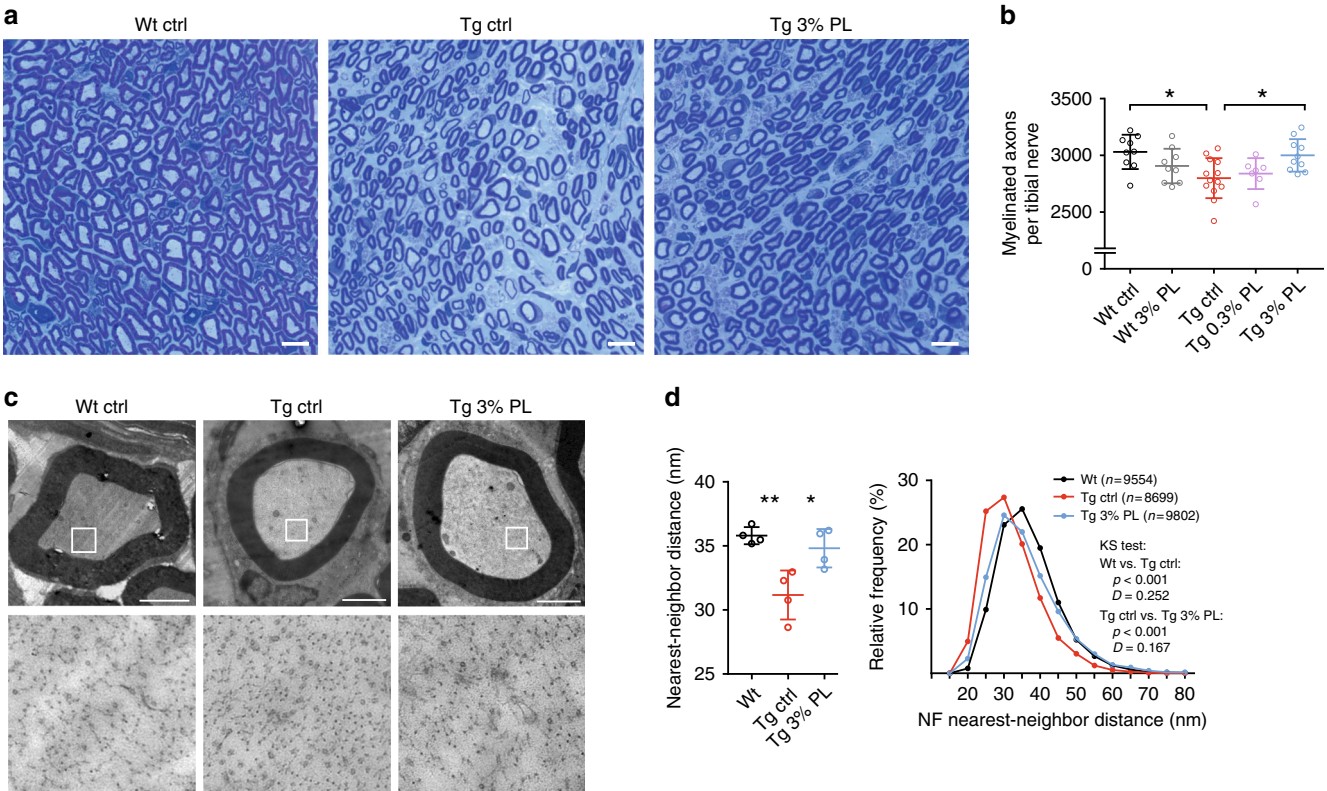

**Fig. 3** Phospholipid therapy in CMT rats improves myelination. **a** Methylene blue semi-thin sections from tibial nerve cross sections at P112 from Wt ctrl (left), Tg ctrl (middle), and Tg 3% PL (right). Note the reduced density of myelinated fibers in Tg ctrl compared with Wt ctrl, which is improved in Tg 3% PL (scale = 10 μm). **b** Light microscopic quantification of **a** displays increased number of myelinated fibers at P112 after 3% PL, but not 0.3% PL treatment (Wt ctrl, black, n = 9; Wt 3% PL, gray, n = 9; Tg ctrl, red, n = 13; Tg 0.3% PL, purple, n = 7; Tg 3% PL, blue, n = 10, one-way ANOVA, Tukey's post test). **c** Representative electron micrographs showing neurofilament densities in axons wt, tg, and tg 3% PL. Whole myelinated fibers (top row, scale 2 μm) with blow-ups (bottom row, scale 100 nm) are shown. **d** Quantification of (**c**) with a nearest-neighbor analysis demonstrating a decreased neurofilament density in tg animals, which is improved upon PL treatment (left panel: n = 4 per group, 20 fibers per animal, quantification of at least 200 neurofilaments per fiber, one-way ANOVA, Sidak's post test, scale = 50 nm, standard deviation (SD), and standard error of mean (SEM); right panel: frequency distribution analyses of all measured neurofilament distances per group, Kolmogorov–Smirnov (KS) test, p value *<0.05 and **<0.01 and ***<0.001)

classes in purified myelin was barely altered between wild-type and CMT rats, with plasmalogens (down in CMT rats) and triacylglycerols (up in CMT rats) as the only deregulated lipids in wild-type versus CMT rat myelin (Supplementary Fig. 4B). However, in purified CMT rat myelin, the proportion of 36 acyl carbon molecules among glycerolipids was decreased and again increased after PL treatment (Fig. 5d). We note that the vast majority of supplemented phospholipid species contained a total of 36 acyl carbons on two fatty acid residues, as the diet comprised of 80% 18-carbon fatty acids (see Methods). The increase of 36 acyl carbon lipids after PL therapy strongly suggests that dietary lipids have been efficiently incorporated into the myelin sheath (Fig. 5d).

**Lipid therapy is effective at different CMT1A disease stages.** We previously demonstrated that treatment of CMT rats with the recombinant growth factor neuregulin-1 early postnatally (P6–P18) significantly improves CMT1A disease pathology[8]. Importantly, early short-term treatment (12 days) was sufficient to improve the disease phenotype over 3 months, whereas neuregulin-1 treatment starting at later time points was less effective[8]. We therefore tested whether the therapeutic effect of PL is also restricted to a specific time window. We treated newborn CMT rats (including their mothers, as lactating pups start feeding solid food only around P15) with 3% PL from P2 to P21,

i.e., during the phase of active myelination (Fig. 6a). Indeed, without altering the body weight, this early short-term therapy was sufficient to improve motor performance when CMT rats were phenotyped at age P21 (Fig. 6b, c). We also observed a significant increase in the number of myelinated fibers in PL-treated CMT rats at that age (Fig. 6d, e), demonstrating that early short-term treatment improves developmental myelination. Next, we asked whether the early postnatal treatment would result in a long-term therapeutic effect. Surprisingly, when short-term-treated CMT rats (until P21) were analyzed after treatment cessation, the improved motor phenotype was still obvious at P30, but disappeared when retested at P50 and P75 (Fig. 6f). Is a continuous PL treatment required to maintain the therapeutic effect? Of note, while the continuous PL treatment from P2 to P112 proved successful on histological and phenotypical level (see above Figs. 2, 3), the analysis of the molecular Schwann cell phenotype in P2–P112 PL-treated CMT rats did not show an improved Schwann cell differentiation or dedifferentiation neither on the level of gene transcription (Supplementary Fig. 5A) nor on the level of MEK/ERK activity (Supplementary Fig. 5B), a pathway crucially involved in Schwann cell dedifferentiation[32]. Another established mediator of Schwann cell dedifferentiation is macrophage-driven inflammation[33]. Corroborating previous studies in various CMT mouse models[34], the rat mutants indeed displayed an elevated number of endoneurial macrophages; this elevation was not changed by the PL treatment at P112

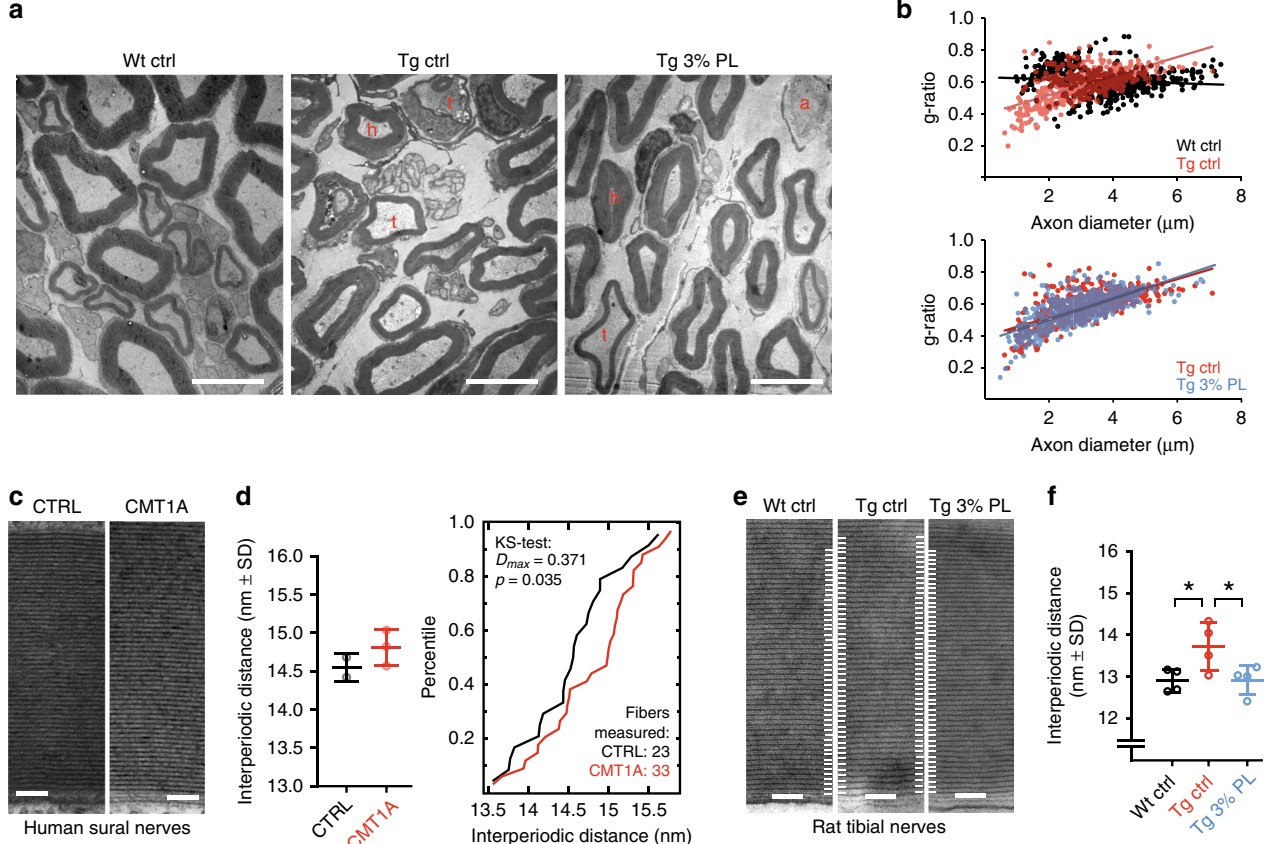

**Fig. 4** Phospholipid therapy in CMT rats improves perturbed myelin ultrastructure. **a** Representative electron micrographs of tibial nerve cross sections from a Wt ctrl (left), Tg ctrl (middle), and Tg rat treated with 3% PL (right) at P112. CMT rats typically display numerous hypermyelinated (h), thinly myelinated (t), and amyelinated (a) fibers. Scale bar: 5 µm **b** Electron microscopic analysis of the myelin sheath thickness by g-ratio measurement shows a steepening in the regression line in Tg ctrl rats (red, $n = 4$), when compared with Wt control rats (black, $n = 5$, upper panel). Treatment with 3% PL (blue, $n = 5$) does not influence perturbed g-ratio distribution (lower panel). Around 82–110 fibers per animal were measured. **c** Ultrastructural analysis of myelin in human control (CTRL) and CMT1A sural nerve biopsies show no gross alteration in myelin morphology in electron micrograph cross sections (scale bar: 100 nm). **d** Quantification of the interperiodic distance from **c** in two control and three CMT1A sural nerve biopsies shows a trend to difference when comparing only the mean distance between CTRL and CMT1A (left panel). A cumulative fraction plot (right panel) including a test for distribution (Kolmogorov–Smirnov test, KS test) of all individually measured myelin sheaths between CTRL (23 measurements) and CMT1A (33 measurements) revealed significant different distribution between the groups, with a higher interperiodic distance in CMT1A. At least 20 adjacent periods were measured for each individual myelin sheath. **e** Ameliorated myelin periodicity in CMT rats after PL treatment. Shown are electron micrographs of Wt and Tg tibial nerve cross sections (30,000×, scale bar 50 nm). Fifty adjacent major dense lines are marked with white ticks. Note that 50 ticks in the Tg ctrl myelin spans more width than in Wt and Tg 3% PL myelin. **f** Quantification of the interperiodic distance from (**e**). Shown is the periodicity at P112, comparing Wt ctrl, Tg ctrl, and Tg 3% PL-treated (P2–P112) rats ($n = 4$ per group, one-way ANOVA, Holm–Sidak's multiple comparison post test). A minimum of 20 periods per fiber in at least 20 fibers per animal were measured

(Supplementary Fig. 5C). Hence, PL treatment, in contrast to neuregulin-1 therapy, does not improve Schwann cell differentiation in the CMT1A rat model, indicating that PL acts as a downstream effector of myelination without affecting the underlying failure of intracellular Schwann cell signaling. Indeed, PL treatment did not ameliorate the reduced PI3K–AKT activity (as measured by AKT phosphorylation) in CMT rat peripheral nerve (Supplementary Fig. 5D). Moreover, while the direct pharmacological activation of PI3K/AKT induces lipid gene transcription (Supplementary Fig. 6A)[35], PL-treated CMT rats showed no rescue of mRNA expression of myelin and lipid genes at P21 (Fig. 7a). In line, we observed no transcriptional upregulation of the rate-limiting enzymes for cholesterol biosynthesis as well as for the major myelin protein *Mpz* in vitro after PC treatment of myelinating co-cultures (Supplementary Fig. 6B). Only, a slight upregulation in wild-type co-cultures was found for *Mbp* mRNA in vitro (Supplementary Fig. 1B). We hence postulate that the promyelinating effect of phospholipids are

downstream of a transcriptional regulation, and that Schwann cells may execute cholesterol and myelin protein biosynthesis via post-transcriptional changes by compensation and regulation on the protein level, e.g., by modulating enzyme activities. In order to demonstrate in proof-of-principle that PC treatment indeed results in an increased biosynthesis of other myelin components, we took advantage of differentiating primary rat Schwann cell monocultures to measure the cholesterol production as a function of PC treatment. Importantly, here we found more cholesterol to be produced in Schwann cells treated with PC, without a concomitant change in cholesterol-related mRNA expression (Fig. 7b, c).

We conclude that PL treatment does not induce intracellular lipid biosynthesis on the transcriptional level, and that the therapeutic effect is mediated by a direct compensation of Schwann cells lipid metabolism. This also explains the observed fading of the therapeutic benefit after PL treatment cessation (Fig. 6f).

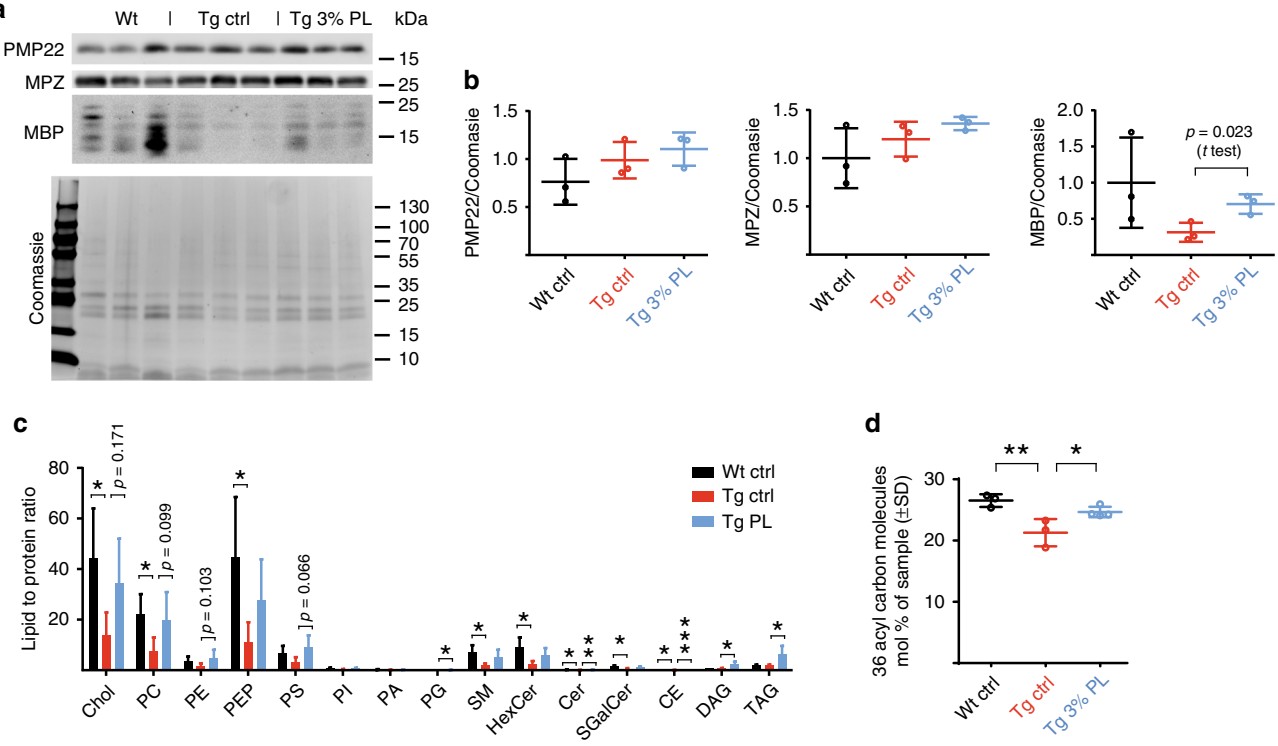

**Fig. 5** Phospholipid therapy in CMT rats improves myelin composition. **a** Western blot analysis of PMP22, MBP, and MPZ from purified P112 sciatic nerve myelin from Wt, Tg ctrl, and Tg 3% PL rats aged 112 days. As loading control, a Coomassie staining of the gel was performed (n = 3 per group). Shown is the Coomassie gel for the PMP22 and MBP blot. MPZ was analyzed on a second blot. **b** Quantification of **a** shows an increased PMP22/MBP ratio in the myelin of Tg compared with Wt controls (left panel, one-way ANOVA). In non-treated (Tg ctrl) versus treated (Tg 3%PL) CMT rats, no quantitative difference of PMP22 in myelin could be detected when normalized to Coomassie (middle panel), whereas more MBP could be detected in the treated group (right panel, Student's t test). **c** Mass spectrometric analysis of sciatic nerve myelin from Wt ctrl (black, n = 4), Tg ctrl (red, n = 3), and Tg 3% PL (blue, n = 3) rats, purified after study end (P112), shows decreased lipid-to-protein ratios for many lipid classes in the Tg ctrl rats, compared with Wt ctrl rats. PL treatment (not significantly) improved lipid-to-protein ratios in purified myelin. Chol cholesterol, PC phosphatidylcholine, PE phosphatidylethanolamine, PEP PE plasmalogens, PS phosphatidylserine, PI phosphatidylinositol, PA phosphatidic acid, PG phosphatidylglycerol, SM sphingomyelin, HexCer hexylceramide, Cer ceramide, SGalCer S-galactosylceramide, CE cerebroside, DAG diacylglycerol, TAG triacylglycerol. Non-adjusted p values are shown (one-way ANOVA). **d** Mass spectrometric analysis of sciatic nerve myelin from Wt ctrl (black, n = 4), Tg ctrl (red, n = 3), and Tg PL (blue, n = 3) rats, purified after study end (P112), displaying only molecules with 36 acyl carbons, which are less abundant in CMT compared with Wt rat myelin and is normalized after PL therapy, suggesting that the supplemented phospholipids (mostly comprised of C18 fatty acids, see Methods) have reached the myelin sheath (one-way ANOVA, Sidak's post test mean ± standard deviation (SD); p value *<0.05 and **<0.01 and ***<0.001)

Accordingly, PL treatment starting after postnatal myelination should still improve CMT1A disease, as lipid biosynthesis in affected Schwann cells remains transcriptionally downregulated throughout life. In order to confirm this hypothesis, we fed the CMT rats with chow enriched with 3% PL, now from P21 until P90 (Fig. 7d). Indeed, a longitudinal analysis of the motor phenotype revealed an improved motor performance when tested after 2 months of treatment (Fig. 7e). Importantly, at study end, treated CMT rats showed muscular preservation (Fig. 7f), improved axonal preservation, as measured by improved CMAP, despite unaltered NCV (Fig. 7g), and a significantly increased number of myelinated fibers in the tibial nerve (Fig. 7h) compared with non-treated controls. In summary, we have demonstrated that PL application directly improves the function of myelinating Schwann cells, and that treating *Pmp22* transgenic rats with PL improves CMT1A disease in this model at the behavioral, physiological, and histological level.

## Discussion

CMT1A is characterized by dysmyelination during early postnatal development along with a slowly progressive demyelination and axonal loss in adult peripheral nerves. Using a *Pmp22* transgenic disease model, we here demonstrate that mutant Schwann cells are characterized by an endogenous defect of efficient lipid biosynthesis that originates at the transcriptional level, coincides with the onset of myelination, and persists throughout life. Importantly, lipid gene transcription has been recently shown to be controlled by the PI3K/AKT/mTOR signaling cascade in Schwann cells[35,36]. In line, we previously found that *Pmp22* transgenic Schwann cells are characterized by a strongly decreased PI3K/AKT signaling pathway during postnatal development[8]. Hence, the defective lipid gene transcription in CMT1A may be the direct result of reduced AKT/mTOR activation in CMT1A. In line with this hypothesis, we were indeed able to induce the transcription of lipid genes by stimulating the PI3K/AKT pathway in *Pmp22*-overexpressing Schwann cells in vitro. In turn, we did not find enhanced PI3K/AKT activity in PL-treated CMT rats in vivo, and reduced lipid gene transcription was not ameliorated in PL-treated CMT1A rats. Also, molecular Schwann cell differentiation was not rescued by PL therapy in CMT1A animals. Hence, we suggest that impaired lipid biosynthesis is downstream of impaired PI3K/AKT signaling, which eventually hinders myelin biosynthesis by *Pmp22*-overexpressing Schwann cells.

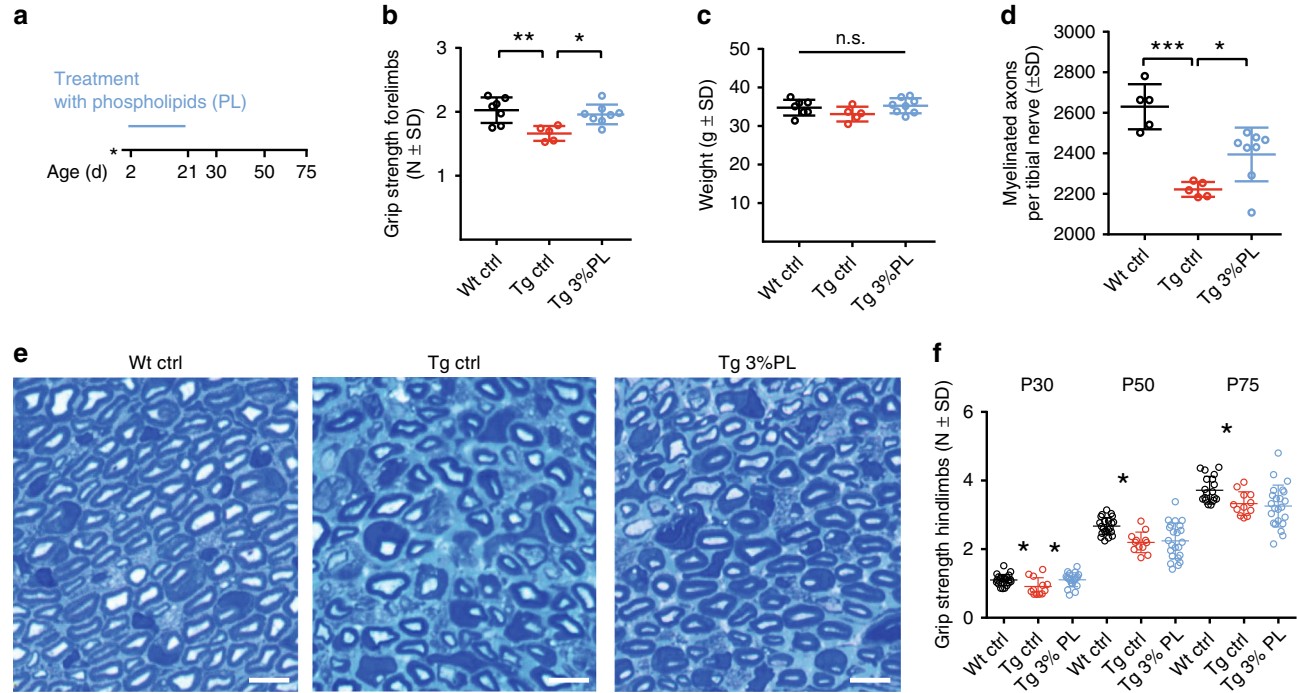

**Fig. 6** Early phospholipid therapy ameliorates neuropathy in CMT rats. **a** CMT rats were fed with either normal diet or a food enriched with 3% phospholipids (PL, see Methods section for details) from postnatal day (P) 2 to P21 (early short term, Figs. 6b–f, 7a) and analyzed at P21 (**b–e**) or at P30, P50, and P75 (**f**). **b**, **c** One cohort of early short-term-treated rats was analyzed at P21. Tg ctrl rats (red, $n = 5$) display a reduced grip strength (**b**) compared with Wt controls (black, $n = 7$). Treatment with 3% PL from P2 to P21 improves grip strength in CMT rats (blue, $n = 8$) (left panel). Body weight measurement reveals no overall difference (**c**). One-way ANOVA, Sidak's post test. **d**, **e** Light microscopic examples (**e**) and quantification (**d**) of the number of myelinated fibers per tibial nerve cross section at P21. CMT rats (red, $n = 5$) display less myelinated fibers when compared to Wt controls (black, $n = 5$). Treatment of CMT rats with 3% PL (blue, $n = 8$) significantly improves the number of myelinated fibers (one-way ANOVA, Tukey's post test, scale = 10 μm). **f** A second cohort of rats treated with the early short-term paradigm (P2–P21) was analyzed for grip strength at P30, P50, and P75. Whereas the treatment effect was still visible at P30, no effect could be seen anymore at P50 and P75 (Wt ctrl, black, $n = 22$; Tg ctrl, red, $n = 12$; Tg 3% PL, blue, $n = 21$). One-way ANOVA, Tukey's post test, $p$ value: *<0.05, **<0.01, ***<0.001, n.s., not significant

The persistent defect of Schwann cell differentiation and decreased PI3K/AKT signaling may explain why the cessation of PL treatment is accompanied by a fading of the therapeutic effect. Indeed, we found lipid transcription to be still downregulated in adult *Pmp22* transgenic Schwann cells compared with wild-type controls, suggesting that also the (in general lower) lipid need in adulthood cannot be fulfilled by diseased Schwann cells in CMT1A. The previously described slowed and hence prolonged myelination in CMT1A[8], together with ongoing de- and remyelination in adult disease stages is likely to require a continuous lipid biosynthesis by *Pmp22* transgenic Schwann cells, next to physiological myelin turnover.

We hence hypothesized that targeting a downstream problem, the reduced availability of synthesized lipids, should ameliorate the cellular defect of CMT1A animals with regard to myelin sheath production. Indeed, when we applied fluorescence-labeled exogenous lipids, the added lipids were utilized by Schwann cells and incorporated into the myelin sheath, which resulted in strongly improved myelination compared with non-treated controls. Of note, in vitro phosphatidylcholine treatment induced the production of cholesterol by *Pmp22* transgenic Schwann cells without concomitant changes in lipid transcription, suggesting that post-transcriptional mechanisms such as compensation and regulation of enzyme activities enables Schwann cells to execute lipid and myelin protein biosynthesis upon phospholipid supplementation.

Importantly, when we translated these findings into a therapeutic trial in vivo, we detected improved myelin biosynthesis along with an amelioration of the neuropathic phenotype in *Pmp22* transgenic rats based on a simple PL-enriched diet. We note that in vivo PL treatment did not alter the number of myelinated axons in wild types, in contrast to the in vitro experiments which were performed under starvation in the lipoprotein-deficient medium, whereas this is not the case in vivo, suggesting that the exogenous lipids are not rate limiting in vivo in the wild-type situation under normal chow conditions.

Where does the increased number of myelinated fibers after in vivo PL treatment derive from? We previously demonstrated that CMT1A rats never achieve a normal number of myelinated axons during postnatal development, with a significant difference in the number of myelinated axons from postnatal day 18 on[8]. Those fibers that should be myelinated (caliber >1 μm), but remain amyelinated during development survive within the nerve until later disease stages in adulthood and slowly degenerate with disease progression. Hence, we suggest that PL treatment helps Schwann cells to physiologically myelinate this pool of amyelinated fibers (>1 μm), thereby protecting these fibers against degeneration. Moreover, PL treatment improved the lipid-to-protein ratio in myelin membranes as well as the widened ultrastructural periodicity of the myelin layers in CMT rats. An altered myelin periodicity has also been detected in nerve biopsies from patients with CMT1A[31] (Fig. 4c, d), but the underlying mechanisms were unknown. We detected only minor changes in protein stoichiometry in CMT rats, suggesting that myelin protein composition is not the cause of the altered myelin periodicity in CMT1A. However, a widened myelin periodicity has been

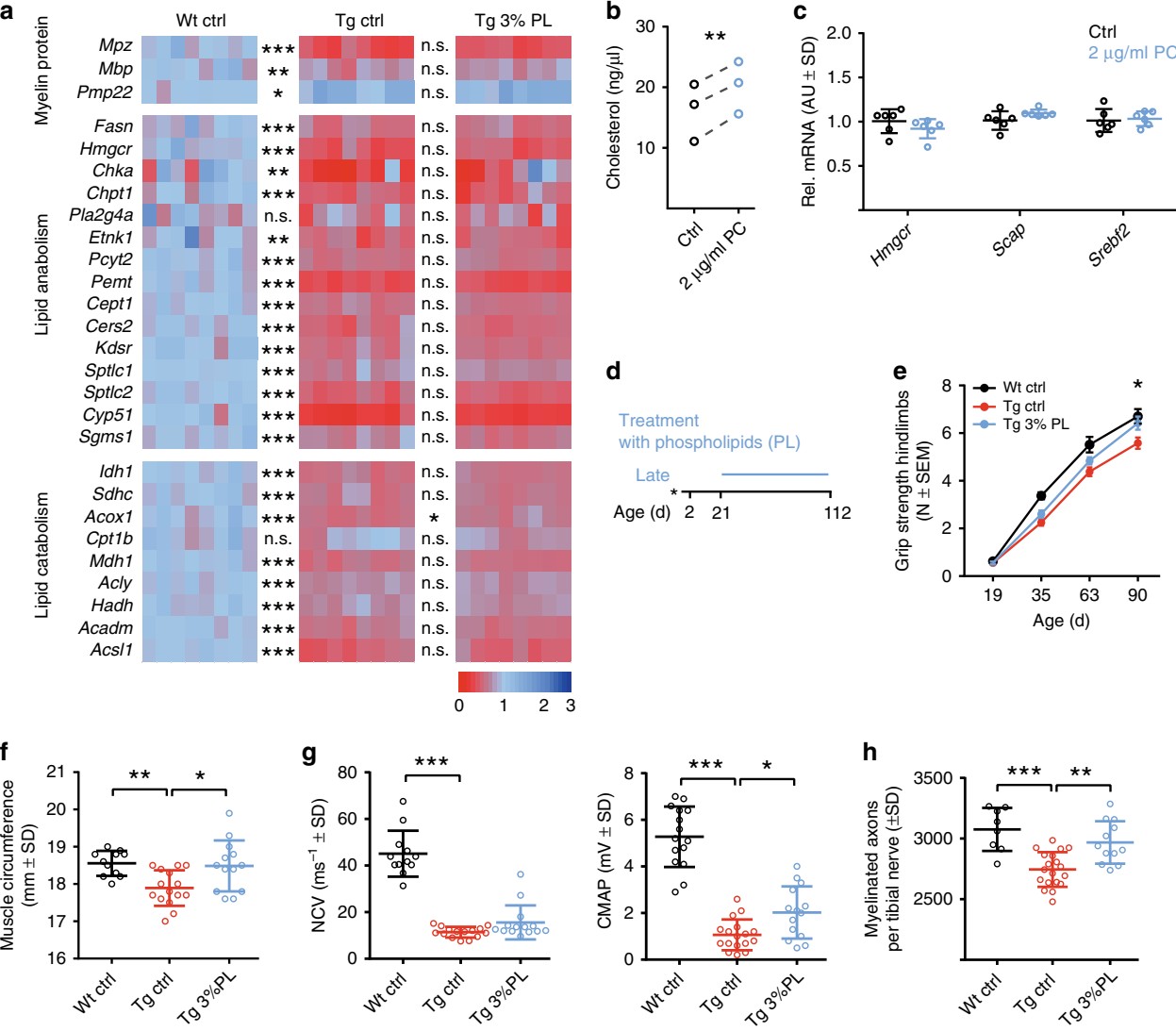

**Fig. 7** Late phospholipid therapy ameliorates neuropathy in CMT rats. **a** Large-scale qPCR analysis in P21 sciatic nerves mRNA extracts of rats that were treated with or without 3% PL, as in Fig. 6a. Analysis of genes encoding myelin, lipid anabolism, and lipid catabolism proteins revealed a consistent downregulation of virtually all respective gene transcripts in Tg animals, compared with Wt controls, which was unaltered upon 3% PL treatment ($n = 8$ per group, one-way ANOVA, Tukey's post test). **b**, **c** Treatment of differentiating Schwann cell monocultures kept in delipidated serum with phosphatidylcholine (PC, 2 µg/ml) revealed an increased cholesterol biosynthesis (**b**, paired analyses, three biological replicates with each two treated versus two non-treated cultures, paired $t$ test) without a concomitant transcriptional change in key enzymes of cholesterol synthesis (**c**, six treated versus six non-treated cultures, Student's $t$ test). **d** A third cohort of rats underwent a late long-term treatment paradigm from P21 on. **e** After 2 months of treatment, at P90, CMT control rats (red, $n = 20$) performed worse when compared with wild-type controls (black, $n = 12$), whereas CMT rats treated with 3% PL (blue, $n = 15$) showed significant improvement in grip strength. No phenotype improvement was observed at earlier time points (P35 and P63). One-way ANOVA, Tukey's post test. **f** At the end of the late long-term study (at P112), CMT rats displayed a rescue of the reduced muscle circumference of the lower forelimbs (Wt ctrl, black, $n = 11$; Tg ctrl, red, $n = 15$, Tg 3% PL, blue, $n = 13$). One-way ANOVA, Tukey's post test. **g** Electrophysiological recordings on the tail motor nerve of late long-term-treated rats at P112 showed no effect on reduced nerve conduction velocity (NCV; left panel), but an increase in the compound muscle action potential (CMAP) amplitudes (right panel; Wt ctrl, black, $n = 12$; Tg ctrl, red, $n = 15$; Tg 3% PL, blue, $n = 14$). **h** At the end of the late long-term study (at P112), CMT rats displayed a rescue of the reduced number of myelinated fibers per tibial nerve cross section (light microscopic quantification; Wt ctrl, black, $n = 8$; Tg ctrl, red, $n = 19$, Tg 3% PL, blue, $n = 12$). One-way ANOVA, Tukey's post test, $p$ value: *<0.05, **<0.01, ***<0.001; mean ± standard deviation (SD) or standard error of mean (SEM)

recently demonstrated in mouse mutants in which fatty acid synthesis or plasmalogen phospholipid synthesis had been ablated[16,37], indicating that decreased amount of lipids in myelin of *Pmp22* transgenic Schwann cells (Fig. 5c) could be the direct cause for the observed ultrastructural changes. That lipid supplementation can indeed affect myelin composition has been suggested by pulse chase experiments, showing that intraperitoneally injected phospholipid precursors are rapidly taken up by

Schwann cells and deposited into the myelin[38], and further demonstrated in our present findings after intravenous injection of fluorescently labeled PC. Hence, myelin lipid composition in the periphery is plastic and can be influenced by both exogenous and endogenous availability of lipids.

Whether the improved myelin periodicity may ultimately contribute to the observed increase of nerve conduction cannot be unequivocally deciphered, though it may be plausible

that an optimized molecular composition and architecture of myelin would reduce the capacitance and increase the resistance in nerve fibers and, hence, improve nerve conduction[39]. Of note, we did not observe changes in other parameters involved in nerve conduction speed, such as myelin sheath thickness, internodal length, and axonal caliber (Fig. 4a, b and Supplementary Fig. 3A, D).

Importantly, we observed a therapeutic effect of PL treatment not only during early postnatal development, but also when we started treatment in advanced disease stages in adult CMT rats. This finding is of major relevance in view of a potential translation of our findings into clinical trials with patients affected by CMT1A disease. Indeed, although the first symptoms occur already during childhood, most patients seek medical advice and respective therapeutic options only in young adulthood. Moreover, as CMT1A disease is characterized by strong clinical variability, a therapeutic intervention in adult, symptomatic patients can be more easily adapted to disease burden and prevent overtreatment and side effects in pediatric patients. Importantly, dietary phospholipids have been tested in many clinical trials and showed no substantial side effects[40]. In clinical practice, phospholipid treatment has even been associated with a lower cardiovascular risk, anti-inflammatory effects in rheumatoid arthritis, and a potential positive influence on memory and cognition in neurological disorders[40]. We hence suggest that a phospholipid therapy constitutes a promising translatable therapeutic rationale for CMT1A disease, which may also be applicable in conjunction with other newly emerging therapeutic options, such as treatment with antisense oligonucleotides suppressing *Pmp22* mRNA, which has recently been shown to improve the disease phenotype in CMT1A animal models[41].

In conclusion, we have identified perturbed lipid metabolism as a disease mechanism downstream of *Pmp22* duplication in CMT1A, and found that dietary lipid supplementation acts as a downstream effector of Schwann cell function, which bypasses the inefficient expression of genes for lipid synthesis in *Pmp22* transgenic Schwann cells. This improves myelin biosynthesis and the neuropathic phenotype of a CMT1A rat model, demonstrating that lipid supplementation should be considered as a new therapeutic approach to CMT1A disease.

## Methods

**Transgenic rats and mice**. *Pmp22* transgenic rats[9], (SD-Tg(Pmp22)Kan), were used for experimental therapy trials, while E13.5 *Pmp22* transgenic mice, (Tg (PMP22)C61Clh), embryos were used for in vitro experiments[42]. For genotyping PCR, genomic DNA was extracted from the tail biopsies using nexttec Kit according to the manufacturer's procedures. For routine genotyping, we used the following PCR primers in a coamplification reaction. Primer sequences: Genotyping primers: For *Pmp22* transgenic rats, sense 5′-CCAGAAAGCCA GGGAACTC-3′, and antisense 5′-GACAAACCCCAGACAGTTG-3′, and for *Pmp22* transgenic mice, sense 5′-TCAGGATATCTATCTGATTCTC-3′ and antisense 5′-AAGCTCATGGAGCACAAAACC-3′. All animal experiments were conducted according to the Lower Saxony State regulations for animal experimentation in Germany, as approved by the Niedersächsische Landesamt für Verbraucherschutz und Lebensmittelsicherheit (LAVES), and in compliance with the guidelines of the Max Planck Institute of Experimental Medicine.

Inclusion and exclusion criteria were pre-established. Only male rats were used for the therapy trials. Animals were randomly included according to the genotyping results, age, and weight into the experiments. Animals were excluded prior to experiments in case of impaired health condition or weight difference of more than 10% with the average group. Exclusion criteria, during or after the experiment was performed, comprise impaired health condition of individual animals not attributed to genotype or experiment (according to veterinary), or weight loss >10% of the average group. No animals had to be excluded due to illness/weight loss in all performed animal experiments. Exclusion criteria regarding the outcome assessment were determined with an appropriate statistical test, the Grubbs' test (or ESD method), using the statistic software GraphPad (Prism).

Animal experiments (phenotype analyses, electrophysiology and histology) were conducted in a single blinded fashion towards the investigator. Selection of

animal samples out of different experimental groups for molecular biology/ histology/biochemistry was performed randomly and in a blinded fashion.

**Phospholipid therapy**. Standard rat chow was used (PS R-Z, Ssniff Spezialdiäten GmbH), according to the supplier, consisting of 35% starch, 21.2% crude protein, 6.7% crude ash, 5% sugar, 4.4% crude fiber, and 3.8% crude fat. In the phospholipid treatment study, soy phospholipids (SIGMA, P3644; 55% phosphatidylcholine, 20% phosphatidylethanolamine) was mixed to the standard chow at two different concentrations, 0.3 and 3%. The fatty acid contents of the supplemented phospholipids were ~60% C18:2 (linoleic), 17% C16:0 (palmitic), 9% C18:1 (oleic), 7% C18:3 (linolenic), and 4% C18:0 (stearic), with other fatty acids being minor contributors.

The animals, wild types and *Pmp22* transgenics, were fed with 0.3 or 3% lecithin-chow in four different therapeutic paradigms. (1) Long-term treatment from P2 to P112, (2) early short-term treatment from P2 to P21, and (3) early short-term treatment late effect, in which the animals were fed with 3% lecithin-chow from P2 to P21 and then the animals were switched to normal food until P80. The last feeding paradigm (4) was the late long-term treatment, in which the animals received chow mixed with lecithin from P21 to P90. We performed a priori power analysis integrating the disease variability in order to calculate the required group sizes. In addition, we have stratified all CMT rats for the late long-term treatment (P21–P90) before therapy start (P19). However, for the trials starting right after birth (P2–P21 and P2–P90), stratification was not possible (as phenotype tests at that age are not applicable). In these trials, rats were randomly allocated to the different treatment groups. At each treatment paradigm's termination time point, motor phenotyping and electrophysiology of the animals were performed and finally, tissues were collected after killing the animals for further histological and molecular analysis.

In order to test the incorporation of circulating phosphatidylcholine into the myelin sheath, 500 μg BODIPY-labeled TopFluor PC (Avanti polar lipids, #810281) dissolved in 25 μl of pure ethanol was infused into the tail vein of a 15-day-old CMT rat. One week later, the rat was killed and the tissue was collected for analysis.

**Motor phenotyping**. The impact of lecithin treatment on the rats forelimbs' and the hindlimbs' grip strength was examined by standardized grip strength tests[22,43]. For hindlimb measurements, forelimbs of the animal were supported and the animal's tail was pulled against a horizontal T-bar (width 14 cm, diameter 3.2 mm) connected to a gauge. In case of forelimb measurements, the animal gripped the same horizontal T-bar during pulling it away from the bar with increasing force. The maximum force (measured in Newton) exerted onto the T-bar before the animal lost grip was recorded, and a mean of at least eight repeated measurements was calculated. All phenotyping analyses were performed by the same investigator who was blinded toward the genotype and treatment arm. To assess the extent of muscle mass, the skin of the left forelimb was removed and the muscle circumference was measured by wrapping a non-sterile silk suture thread (0.65 mm; F.S.T. cat#18020-03) five times around the group of muscles (adjacent non overlying), attached to the radius and the ulna, starting from the joint toward the paw. The wrapped thread length was measured to the nearest millimeter using a normal desk ruler (adapted from ref. [44]).

**Electrophysiology**. Nerve conduction velocities (NCVs) and compound muscle action potentials (CMAPs) were measured[22,43]. Briefly, rats were anesthetized with xylazinhydrochloride/ketaminhydrochloride (8 mg per kg body weight/100 mg per kg body weight). For distal stimulation, two steel electrodes (Schuler Medizintechnik, Freiburg, Germany) were placed along the tibial nerve above the ankle and for proximal stimulation other two steel electrodes were placed at the sciatic nerve notch. Recording needle electrodes were inserted into the plantar muscles. Supramaximal square wave pulses (100 ms duration) were applied using a Toennies Neuroscreen (Jaeger, Hoechsberg, Germany). NCVs were calculated using the distance between the proximal and distal stimulation electrodes, while the leg is completely extended, and sciatic nerve conduction latency measurements.

**Histology**. For light microscopy: sciatic nerves of the rats were kept in 2.5% glutaraldehyde and 4% paraformaldehyde in 1× phosphate buffer for 1 week. Afterward, probes were osmicated and embedded in epoxy resin (Serva). Semi-thin sections (0.5 μm) were prepared (Leica RM 2155, using a diamond knife Histo HI 4317, Diatome) and stained with a mixture of 1% toluidine blue and azur II–methylene blue for 1 min at 60 °C. Microscopic images were collected using a ×100 lens (Leica DMRXA), and digital images were obtained using Axiophot microscope (Zeiss) equipped with AxioCam MRC (Zeiss) and zen 2012 software. Counting of myelinated axons was carried out on whole sciatic nerve cross sections, manually using ImageJ (NIH).

For electron microscopy: Ultrathin (50–70 nm) sciatic nerve cross sections were treated with 1% uranylacetate solution and lead citrate and analyzed using a Zeiss EM10 or EM109 (Leo). Myelin sheath thickness comparison was carried out via calculation of the g-ratio, which is a numerical ratio between the fiber's diameter and the diameter of the same fiber and the myelin sheath wrapping it. For g-ratio comparison at least 150 fibers were randomly analyzed from each sciatic nerve at

×3000 magnification. Ultrastructural analysis was performed by measuring the periodicity, i.e., the distance between two adjacent major dense lines, for at least 20 periods per myelinated fiber of at least 10 (human sural nerves) and 20 (rat tibial nerves) myelinated fibers per sample at a ×30,000 magnification. For all histological quantification, the experimenter was blinded with respect to genotype and treatment of all animals.

The morphometric analysis of human sural nerve biopsies was performed in three patients affected by CMT1A disease (all three with a genetically confirmed duplication of the *PMP22* gene, two males, one female, age 59, 60, and 44 years, respectively) and on two biopsies of individuals without signs and microscopic characteristics of a peripheral neuropathy (both females age 39 and 51 years).

**Cell culture.** Dorsal root ganglia (DRGs) cultures were prepared by isolating DRGs from wild-type or *Pmp22* transgenic mouse embryos (C61 line[42] at embryonic day 13.5 (E13.5) and plated according to standard procedure[45]. Dissociated DRGs, with trypsin, were plated at a density of $10 \times 10^4$ cells per 1 mm coverslip coated with collagen (Gibco). The cells were kept in a growth medium, 10% deactivated Hyclone fetal bovine serum (GE healthcare & Life technologies) and 50 ng/ml NGF (alomone labs) in a minimum essential medium (MEM), (Gibco) for 1 week. In order to induce myelination, growth medium was supplemented with 50 ng/ml ascorbic acid (AA) (SIGMA) every other day. In order to test the effect of exogenously applied phosphatidylcholine (PC) on myelination in vitro, the cells were switched to myelination medium composed of MEM containing 10% lipoprotein-deficient serum (LPDS) (Sigma), 50 ng/ml NGF, 50 ng/ml ascorbic acid (AA) after growing the cells for 1 week in normal growth medium. The cultures grown on LPDS were supplemented with 2 μg/ml PC (Avanti polar lipids, #441601 G) dissolved in ethanol while control cultures received equal amounts of ethanol as PC-treated cultures. For testing the direct incorporation of PC into myelin, by the time of myelination induction the cells were treated with 2 μg/ml BODIPY-labeled TopFluor PC (Avanti polar lipids, #810281), whereas control cultures were treated with 2 μg/ml BODIPY-labeled pentanoic acid (ThermoFisher, #D3834).

Rat Schwann cells were prepared from sciatic nerves of four newborn rats (4-day old)[46]. For cell expansion, media were supplied with 10 ng per ml medium recombinant human neuregulin-1 EGF-like domain (rhNRG1, Reprokine) and 4 μM forskolin (SIGMA). Cells were deprived of rhNRG1 for 1 week before freezing and storage. For experiments, independent Schwann cell preparations (as biological replicates) were defrosted, replated, and cultured on resting medium (DMEM + 10% FCS) for 3 days. Before experimental treatment with the specific PI3K activator (740YP, Tocris), Schwann cells were kept on serum-reduced medium (1% FCS) for 1 day. Schwann cells were collected 6 h after treatment began. For the cholesterol assay, Schwann cells were kept on DMEM containing 10% LPDS (Sigma) and differentiated by the addition of 1 mM dbcAMP (Sigma) for 2 days with or without 2 μg/ml phosphatidylcholine. Schwann cells from three independent preparations were used for cholesterol quantification. Gene expression analysis from Schwann cells cultured under the same condition was performed in six replicates.

Cholesterol amounts of primary Schwann cell cultures were measured using a Cholesterol assay kit (Abcam), essentially according to the manufacturer's protocol. In brief, cells where harvested in PBS and extracted in a mixture of chloroform:isopropanol:NP-40 (7:11:0.1). The organic phase was dried under vacuum at room temperature, resuspended in the assay buffer and subjected to a colorimetric reaction. Absorbances were measured in a microplate reader and related to cholesterol concentrations via comparison with a standard curve.

**Immunocytochemistry.** Cells were fixed (4% paraformaldehyde (PFA) in 1× PBS for 10 min), permeabilized (ice-cold methanol 95% and acetone 5% mixture at −20 °C for 4 min) and then incubated for 1 h in blocking solution (4% horse serum, 2% bovine serum albumin (BSA), and 0.1% porcine gelatin. Primary antibodies (polyclonal rabbit anti-MBP (1:400; Dako) and monoclonal mouse anti-class III β tubulin (1:500; Covance)) were diluted in blocking solution and applied at 4 °C overnight. Coverslips were washed three times with 1× PBS, then secondary antibodies (alexa 488 donkey anti-rabbit (1:400) (Invitrogen) and alexa 555 donkey anti-mouse (1:400) (Invitrogen)) diluted in blocking solution containing 0.2 μg/ml 4′,6′-diamidino-2-phenylindole (DAPI) (Sigma), were applied at room temperature for 1 h. Only in case of TopFluor phosphatidylcholine in vitro application a different combination of secondary antibodies was used. Finally, coverslips were washed with PBS, shortly immersed in distilled water and mounted on slides with aqua-polymount (Polysciences). Fluorescence Images were obtained with fluorescence Zeiss Axioskop microscope equipped with MRM camera (Zeiss). Acquisition and processing of images was carried out with Zen2-blue edition (Zeiss), ImageJ (NIH), Photoshop CS (Adobe), and Illustrator 10 (Adobe) software. For the quantification of myelination, the total number of myelin basic protein (MBP)-positive myelin segments on each coverslip was counted, and statistics were done using the two-tailed Student's *t* test.

**Immunohistochemistry.** In order to analyze the incorporation of circulating fluorescent PC into the myelin sheath (see above), sciatic nerves from PC infused rats were cryoembedded and longitudinal sections (20 μm) were prepared. The sections were rehydrated with 0.1 M PBS and fixed with 4% PFA in 0.1 M PBS and washed with PBS. Then permeabilization (0.4% Triton in PBS for 30 min) and blocking (4% horse serum (HS), 0.2% Triton in PBS for 30 min) of the fixed tissues

was carried out. The primary antibody (anti-MBP polyclonal rabbit (1:500; Dako)) was diluted in 1× PBS containing 1% HS, 0.05 % Triton and applied overnight at 4 °C. After washing, the secondary antibody (Alexa 555 anti-rabbit 1:500 (Invitrogen)) and DAPI were diluted in 1.5% HS in PBS and applied at room temperature for 2 h. Finally, the slides were washed with PBS, double distilled water, and covered with aqua-polymount (Polysciences). For staining of activated endoneurial macrophages[47–49], fresh-frozen tibial nerve samples were fixed in acetone, blocked for 30 min with 5% bovine serum albumin (BSA) in 0.1 M PBS, followed by incubation overnight with mouse anti-rat ED1 antibodies (1:500, MAK0341R, Linaris) in 1% BSA in 0.1 M PBS at 4 °C. After washing steps with PBS corresponding Cy3-conjugated secondary antibodies were added for 1 h at RT. Nuclei were visualized by incubation with DAPI (1:500,000, D9542, Sigma-Aldrich) for 10 min at RT. All samples were embedded after a final washing step with Aqua-Poly/Mount® (Polysciences). Digital fluorescence microscopic images were acquired using an Axiophot 2 microscope (Zeiss) equipped with a CCD camera (Visitron Systems) and afterward processed with Photoshop CS3 (Adobe).

**Mass spectrometry of purified myelin and milk.** A myelin-enriched light weight membrane fraction was purified from rats' sciatic nerves homogenized in 0.27M sucrose[50,51]. The protein concentration was measured by Lowry assay using DC protein assay kit (BioRAD) according to the manufacturer's instructions and/or with protein gel silver[52]. The silver gel was imaged with hp Scan jet 6390C (HP intelligent scanning technology) and the density of each lane was measured with ImageJ (NIH).

The amount of 1 μg of myelin membranes per sample were subjected to lipid extractions using an acidic Bligh & Dyer, except from plasmalogens, which were extracted under neutral conditions[53]. Lipid standards were added prior to extractions, using a master mix containing 50 pmol phosphatidylcholine (13:0/13:0, 14:0/14:0, 20:0/20:0; 21:0/21:0, Avanti Polar Lipids) and sphingomyelin (d18:1 with *N*-acylated 15:0, 17:0, 25:0, semi-synthesized as described in ref. [54], 200 pmol D6-cholesterol (Cambridge Isotope Laboratory), 25 pmol phosphatidylinositol (16:0/16:0, 17:0/20:4, Avanti Polar Lipids), phosphatidylethanolamine and phosphatidylserine (both 14:1/14:1, 20:1/20:1, 22:1/22:1, semi-synthesized[54], diacylglycerol (17:0/17:0, Larodan), cholesterol ester (9:0, 19:0, 24:1, Sigma), and triacylglycerol (D5- Mix, LM-6000/D5-17:0,17:1,17:1, Avanti Polar Lipids), 5 pmol ceramide and 20 pmol glucosylceramide (both d18:1 with *N*-acylated 15:0, 17:0, 25:0, semi-synthesized as described[54], 50 pmol SGalCer di18:1/17:0 (Avanti Polar Lipids), 10 pmol phosphatidic acid (17:0/20:4, Avanti Polar) and phoshatidylglycerol (14:1/14:1, 20:1/20:1, 22:1/22:1), semi-synthesized as described[54]. Phosphatidylethanolamine plasmalogen (PE P-)-containing standard mix was supplemented with 22 pmol PE P-Mix 1 (16:0p/15:0, 16:0p/19:0, 16:0p/25:0), 31 pmol PE P- Mix 2 (18:0p/15:0, 18:0p/19:0, 18:0p/25:0), 43 pmol PE P-Mix 3 (18:1p/15:0, 18:1p/19:0, 18:1p/25:0). Semi-synthesis of PE P was performed[55].

Lipid extracts were resuspended in 60 μl methanol and samples were analyzed on an AB SCIEX QTRAP 6500 + mass spectrometer (Sciex, Canada) with chip-based (HD-D ESI Chip, Advion Biosciences, USA) electrospray infusion and ionization via a Triversa Nanomate (Advion Biosciences, Ithaca, USA) as previously described[54]. Resuspended lipid extracts were diluted 1:10 in 96-well plates (Eppendorf twin tec 96, colorless, Sigma, Z651400-25A) prior to measurement.

Measurements were performed in 10 mM ammonium acetate in methanol for the analysis of most lipids except for sulfatides which were measured in 0.005% piperidine in methanol in negative ion mode. Precursor and neutral loss scanning was employed to measure phosphoglycerolipids, sphingolipids, and glycerolipids as described[54]. Sulfatides were analyzed by precursor ion mode selecting for negatively charged fragment ions with *m/z* 97. Remaining samples were subjected to cholesterol determination as described[56]. Data evaluation was done using LipidView (ABSciex) and an in-house-developed software (ShinyLipids). The lipid classes' concentrations defined by mass spectrometric analysis were normalized to the measured protein concentration as assessed by standard Lowry assay (BioRad) and silver gel densitometry. To test for a potential overlap of endogenous lipids and standards we additionally performed lipid extractions in the absence of lipid standards and subjected the samples to MS analysis and data evaluation. Lipid standards used for MS analysis did not significantly overlap with endogenous lipid species (Supplementary Data 1). As an example, the major phosphoglycerolipids species phosphatidylcholine and phosphatidylethanolamine showed an overlap of standards with endogenous species of 0.1% and 0.5%, respectively (Supplementary Data 1). In average, signal intensities of 0.82 ± 0.7% were found to be present in endogenous lipid samples that overlapped with the *m/z* values of standards.

For analyses of nursing rat dam's milk, dams were treated with 3% PL as described above from offspring's age P2–P9. The P9 pups were killed and the stomachs content collected and pooled per litter (five treated dams and four control dams). Samples were transferred to methanol. Lipids obtained by acidic extraction were subjected to fractionation using Discovery DSC-Si SPE tubes (Sigma-Aldrich). Lipids were resuspended in 1 ml CHCl₃ and loaded twice onto SPE tubes, which were equilibrated in CHCl₃. Lipids were then eluted in three fractions. Fraction 1 was eluted with 3 ml CHCl₃, fraction 2 with 3 ml isopropanol:acetone (1:1, v/v), and fraction 3 with 3 ml methanol. Evaporated samples were resuspended in methanol. MS analysis and data evaluation were performed in 10 mM ammonium acetate in methanol as described above.

**Protein analyses.** Rat sciatic nerves were homogenized using a Precellys24 homogenizer (Bertin Instruments) in sucrose lysis buffer (320 mM sucrose, 10 mM Tris base, 1 mM NaHCO$_3$, 1 mM MgCl$_2$, protease inhibitor (cOmplete Mini, EDTA-free, Roche)). For myelin fractions analysis, samples were prepared according to the standard procedure mentioned above. Protein electrophoresis was carried out using precast gradient gels (NuPAGE 4–12% Bis–Tris, Invitrogen). Relative protein concentrations were determined via colloidal Coomassie staining (Imperial stain, BioRad). Western blots were incubated overnight with primary antibodies against p-AKT, AKT, p-ERK, ERK (all polyclonal rabbit; 1:1000, Cell Signaling), anti-PMP22 (polyclonal rabbit; 1:2000; Assay biotech), anti-MBP are (monoclonal mouse; 1:1000, Biolegend), and P0 (monoclonal mouse; 1:5000, kindly provided by H.P. Hartung, Düsseldorf, Germany[57]). Detection was performed using anti-rabbit and anti-mouse-HRP-coupled secondary antibodies, respectively (1:5000, Dianova), Western Lightning Plus-ECL, Enhanced Chemiluminescence Substrate (Perkin Elmer) and a luminescence Imager (Intas Science Imaging).

**RNA analysis.** Total sciatic nerve RNA was extracted with RNeasy Kit (Qiagen), whereas RNA from cell culture was purified using RLT lysis buffer, according to the manufacture's instruction. Concentration and quality (ratio of absorption at 260/280 nm) of RNA samples were determined using the NanoDrop spectrophotometer (ThermoScientific). Integrity of the extracted RNA was determined with the Agilent 2100 Bioanalyser (Agilent Technologies).

For RT-PCR analysis, cDNA was synthesized from total RNA using poly-Thymin and random nonamer primers and Superscript III RNase H reverse transcriptase (Invitrogen). Quantitative real-time PCR was carried out using the Roche LC480 Detection System and SYBR Green Master Mix according to the manufacturer (Applied Biosystems). Reactions were carried out in four replicates. The relative quantity (RQ) of RNA was calculated using LC480 Software (Roche). Results were depicted as histograms (generated by Microsoft-Excel 2003) of normalized RQ values, with mean RQ value in the given control group normalized to 100%. As internal standards, peptidylprolyl isomerase A (*Ppia*) and ribosomal protein, large, P0 (*Rplp0*) were used. PCR primer sequences can be found in the Supplementary Data 2.

**RNA-seq analysis.** Quality control, read alignment, and differentially expressed genes: RNA-sequencing resulted in ~24 million reads per sample (Supplementary Data 3). Quality assessment was based on the raw reads using the FASTQC quality control tool (v0.10.1)[58]. The sequence reads (single-end 50 bp) were aligned to the rat reference genome (rn6) with Bowtie2 (v2.0.2)[59] using RSEM (v1.2.29)[60] with default parameters. First, the rat reference genome was indexed using the Ensembl annotations (v84.6) with rsem-prepare-reference from RSEM software. Next, rsem-calculate-expression was used to align the reads and quantify the gene and isoform abundance. The output of rsem-calculate-expression gives the read count and TPM value (transcripts per million) for each gene and isoform separately.

The gene ontology (GO) gene sets were obtained from the Molecular Signatures Database (MSigDB) from the Broad Institute UC San Diego (http://software. broadinstitute.org/gsea/msigdb/index.jsp). The gene sets were extracted from MSigDB without any further manipulation and the identifiers are given in the figure legend.

**Statistics.** For power analysis, the software G*Power Version 3.1.7. was used. Power analyses were performed before conducting in vivo therapy experiments (a priori). Adequate Power (1 – beta-error) was defined as ≥80% and the alpha error as 5%.

Differential expression analysis was carried out using gene read counts with DESeq2 package[61]. Genes with less than 5 reads (baseMean) were filtered out. Genes with an adjusted *p* value <0.05 were considered to be differentially expressed. Gene ontology enrichment analysis: Gene ontology (GO) analysis was conducted using WebGestalt[62]. An adjusted *p* value <0.1 using the Benjamini–Hochberg method for controlling the false discovery rate was set as significant for GO terms in biological processes. Results were confirmed using DAVID database for downregulated genes at P18. Lipidomics data were analyzed using LIMMA package[63]. Note that NA values were changed to 0. Linear fit was performed using lmFit function for two groups at P18 (Wt and Tg) and three groups for P112 (Wt, Tg_pla, and Tg_Tx3). For the latter, pairwise comparison was performed using contrasts.fit function for Wt-Tg_pla and Tg_pla-Tg_Tx3. We considered an adjusted (P18) and non-adjusted (P112) *p* value <0.05 significant. Unless indicated otherwise, all other data were processed using MS Excel and GraphPad Prism v6.04. The statistical test that was used to analyze the data is indicated in the figure legends, respectively. Briefly, for comparing two group's Student's *t* test was used, for comparing more than two groups one-way ANOVA with appropriate post test was used and for comparing two or more groups for more than one time point (longitudinal analysis) two-way ANOVA with appropriate post test was used and a *p* value <0.05 was considered significant.

**Data availability statement.** All RNA-seq data sets are accessible under GEO accession number GSE115930. All other relevant data are available from the corresponding authors on reasonable request.

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

## Acknowledgements

We are grateful to A. Mott, A. Fahrenholz, T. Durkaya, and C. Maack (MPI of Experimental Medicine) for excellent technical help. We thank T. Pawelz and M. Wehe for excellent animal care taking. We thank M. Ost and G. Saher for helpful discussions. This project is part of the German network on Charcot–Marie–Tooth Disease (CMT-NET, research project R5, 01GM1511C to R.F. and R.S.; R4, 01GM1511F to R.M.; and R6, S3b, 01GM1511D to J.W. and I.K.) funded by the German ministry of education and research (BMBF, Bonn, Germany). B.B. and C.L. were supported by a DFG grant (SFB/TRR83). M.W.S. was supported by the German Ministry of Education and Research (BMBF, CMT-BIO, FKZ: 01ES0812, CMT-NET, FKZ: 01GM1511C, CMT-NRG, ERA-NET'ERARE3', FKZ: 01GM1605) and by the Association Francaise contre Les Myopathies (AFM, Nr: 15037). M.W.S. holds a DFG Heisenberg Professorship (SE 1944/1-1). T.P. was supported by the European Leukodystrophie Society (ELA 2014-020I1 to M.W.S.). K.A.N. is supported by the DFG (SPP1757 and CNMPB) and holds an ERC Advanced Grant.

## Author contributions

R.F. and R.S. designed the study, performed, and supervised experiments, analyzed data and wrote the manuscript; T.A., L.R., T.P. and J.S. planned, performed, and analyzed experimental PL therapy trials; T.A. also performed cell culture, qPCR, and western blot experiments and is the co-first author of the manuscript with R.F.; B.B. and C.L. performed mass spectrometry; W.M. and T.R. performed and supported electron microscopy of animal materials; J.W. and I.K. performed electron microscopy of human samples; D.K. and R.M. analyzed nerve macrophages; V.B., R.U.R. and S.B. performed and analyzed RNA-seq experiments; D.H., V.S. and D.E. performed cell culture qPCR and western blot experiments; R.M., J.W., I.B., W.C.M. and W.B. contributed to the discussions; K.A.N. contributed to the discussions and to the manuscript; R.F., R.S. and M.W.S. supervised the project.

## Additional information

**Competing interests:** The authors declare no competing interests.

