## [Peer review file · Nature Communications]

Reviewers' comments:

Reviewer #1 (Remarks to the Author):

The study by Fledrich et al. investigates a rat model of CMT1A, with a focus on developing new therapeutic approaches to prevent progressive demyelination and axonal loss in the disease. The study is based on previously published findings in the same model demonstrating that Pmp22 transgenic Schwann cells (SCs) display a significant delay in myelination, with many axons remaining completely unmyelinated until adulthood. These fibers degenerate with time, suggesting that enhancing myelination might rescue axon survival.

In the present study, the authors focus on determining how altered SC differentiation is mechanistically linked to reduced myelination in CMT1A, in particular in the rat model of the disease. The authors demonstrate that abnormalities in SC lipid metabolism is a causative factor in the pathogenesis, as myelin lipids are crucially important for myelin membrane growth and stability/integrity.

In this study, Fledrich et al. use an unbiased approach to demonstrate that SCs in the rat model display a developmental defect that includes reduced transcription of genes required for myelin lipid biosynthesis. As a consequence of this defect, lipid incorporation into myelin is severely reduced, leading to an overall imbalance in the stoichiometry of myelin lipid and proteins, and ultrastructural alterations in myelin itself. The authors also find that supplementation of phosphatidylcholine and phosphatidylethanolamine in the diet is sufficient to rescue myelination deficits of SC in vivo. This treatment not only rescues the cellular and the ultrastructural myelin phenotype, but also improves the clinical phenotype.

Overall, this is a very well executed study with an interesting translational potential. The results also identify a clear biochemical/metabolic defect in SCs of the CMT1A rat model, which is novel. The link between genetic and cellular defects, and the myelin and behavioral phenotype is very interesting and important. I have a few suggestions about expanding the analysis presented in the paper in its current form.

1. It would be important to analyze the effects of diet supplementation with myelin lipids on axons and axonal pathology, including ultrastructural and morphological features – e.g. nodes and paranodes.
2. Early and late treatment with myelin lipids. These are not only different in timing but also in duration. The early treatment is from P2 to P21, and the late treatment is from P21 to P90. It would be interesting to determine whether a short-duration late treatment would also be effective. Is the long duration necessary?

Reviewer #2 (Remarks to the Author):

The authors use a transgenic rat overexpressing peripheral myelin protein 22 (Pmp22), which develops Charcot-Marie-Tooth 1A disease, in order to characterize and potentially prevent its neuropathy phenotype. They use a combination of transcriptomic and lipidomic analysis to demonstrate altered lipid metabolism in peripheral nerves of affected animals. Based on these data they design and evaluate the efficiency of phospholipid supplementation therapy. Interestingly, they demonstrate that both long-term and relatively short-term dietary supplementation of phosphatidylcholine and phosphatidylethanolamine partially alleviate the neuropathy phenotype in transgenic animals. The presented data are very interesting with regard to the potential of this approach to be translated into clinical trials. However, the underlying

mechanistic part of the story is somehow less clear.

The transcriptomic data presented in Figure 1 in part overlap with the data previously published by the same group (Figure 2, Fledrich et al., Brain 2012). The differences and/or similarities between the results from the two studies should be discussed. Also, the previous characterization of CMT1A rat model by the authors (Fledrich et al., Brain 2012) revealed a high variability in the severity of the disease, similar to patients with CMT1A. Did the authors consider to stratify affected animals (mildly vs severely affected) in order to potentially decrease variability in the observed phenotypes pre/post treatment?

The observed effect of phosphatidylcholine (PC) in myelinating cultures (Figure 1b-e) is very interesting and could be explored a bit more. Would the same treatment also work in standard media (not media with delipidated serum)? What happens with myelin protein and cholesterol biosynthesis (both presumably needed for myelination) in PC-treated cultures?

The electrophysiological data presented in Figure 2 are difficult to interpret. Nerve conduction velocity (NCV) is only slightly increased by phospholipid therapy in CMT1A rats. However, the number of myelinated axons (which is the contributing factor for NCV) is almost completely normalized. Increase in compound muscle action potential after phospholipid therapy may also indicate changes in neuromuscular junctions in addition to observed changes in muscle – this aspect could be evaluated.

As described by the authors, the affected animals present a mix of hypomyelinating, amyelinating and hypermyelinating axons (Figure 3). Also, it was previously shown that there is a more pronounced defect in ventral roots (Sereda et al., Neuron 1996). The characterization of morphological changes in Figure 3a-c could therefore be more informative if done separately on roots (in particular ventral roots, containing motor fibers). Also, a more detailed quantification of different abnormalities could be done (e.g. number of unmyelinated axons, number of onion bulbs) since the heterogeneity of the various abnormalities in myelinating Schwann cells may obscure presented g-ratio data.

How does the observed improvement in myelin interperiodic distance in animals treated with phospholipids (Figure 3d-g) connect to concomitant electrophysiological and behavioral changes?

There is an increase in the number of myelinated axons in CMT1A animals on phospholipid therapy (Figures 2g and 4c). From where are these fibers coming? Also, there is no improvement in decreased transcription of lipid genes in CMT1A animals (Figure 4g). While the phospholipid therapy may provide sufficient amounts of phosphatidylcholine and phosphatidylethanolamine for additional myelin, it is unclear from where cholesterol and myelin proteins are coming, which are the main components of the myelin membrane.

Minor comments:

In Figure 1f, please present the images with the same magnification as in 1b in order to have a more global view of myelin segments.

The co-labeling between MBP and BODIPY-PC (Figure 2a) could be done on teased fibers or on nerve cross sections in order to improve resolution.

Reviewer #3 (Remarks to the Author):

The paper by Fledrich et al. describes changes in lipid composition and synthesis in Charcot-Marie-

Tooth type 1A, caused by increased gene dosage of PMP22. The paper demonstrates that these changes are the down stream consequence of changes in transcription levels of the lipid biosynthetic enzymes. The paper goes on to demonstrate that exogenously applied lipids can improve myelination in vitro, and can incorporate into myelin sheaths in vivo after IV injection. The therapeutic potential of this approach is examined in vivo, and modest but significant improvements in myelin and motor performance are described. The time course of these improvements is also examined, and it is determined that the lipid dosing must be maintained to maintain the improvement, but that some improvement can be found even with dosing beginning after development. Most of these experiments were done using a very good rat transgenic model of CMT1A.

The paper is well written and the results are interesting and supported by the data presented. The rationale for the in vivo experiments is well established by the transcriptome and in vitro studies. The in vivo studies use rigorous study design, with randomization of animals into treatment groups, pre-established inclusion and exclusion criteria, and single blind collection of the data. The results are generally well presented (but see points below) and the outcomes are positive, but generally not overstated. The assessment of CMAP amplitudes and myelinated axon numbers are the most convincing data. The straightforward nature of the proposed intervention is appealing.

Addressing the comments below would further improve the manuscript.

The g-ratio results shown in figure 3C are problematic as presented. The relationships between axon size and myelin thickness shown in Figure 3B are the more appropriate indicators of what is happening, and also indicate that PL treatment did not change this relationship. The average g-ratio shown in 3C is just an indirect indicator that there are more small (hypermyelinated) axons than large (hypomyelinated) axons. Panel 3C should simply be removed.

In Figure 3F, a picture of the myelin packing of a PL treated nerve would improve the figure.

The age at which myelinated axon number first becomes significantly lower than control is an important point and should be reported. This matters for interpreting the improvements in myelinated axon number with treatment starting at P21 (Figure 4L).

The improvements in muscle size are problematic. This is particularly true in Figure 4I, where wild type rats treated with PL are not shown, but also in 2D, where the magnitude of the effect of PL on wild type muscles seems comparable to the improvement seen in mutant muscles, but without the statistical significance. The wild type with 3% PL treatment appears to be the result with the lowest number of animals in 2D.

Are phospholipids passed in the milk of nursing dams? The authors acknowledge that rat pups do not eat solid food until P15, which is important for interpreting the results, especially in figure 4, but whether ingestion by the mother is effectively passed to the pups should be noted.

The Discussion is well written and generally ties the results into the larger context well, especially around previous work in the mTOR/AKT pathway. However, the authors should also include a brief discussion of their findings in comparison to the recent JCI paper by Zhao et al., showing that downregulation of PMP22 transcript levels using a gene therapy approach is also beneficial. Key points seem to be the post-onset benefits seen by Zhao in comparison to those reported here, and the possibility of combining these approaches.

Minor points:

A reference for the PMP22 transgenic mouse strain used for the Schwann cell cultures should be provided in the methods section.

Reviewer #4 (Remarks to the Author):

The manuscript titled "Targeting myelin lipid metabolism as an effective therapeutic strategy in a rodent model of CMT1A neuropathy" by Fledrich et. al describes the role of lipid metabolism in Charcot-Marie Tooth Disease 1A. Using "omics" approach the authors showed a decrease in lipid metabolism gene expression as well as lipid content in myelin of pmp22 transgenic rats. Besides providing a plausible missing link in CMT1A pathogenesis, this results offer a therapeutic target in treating CMT1A. Using an easily translatable therapeutic approach, phospholipid supplementation, the authors showed improvement of CMT1A. This manuscript provides yet another evidence to the importance of lipids in the nervous system. Also, this manuscript showed how an omics data can be translated into a potent therapeutic application. The authors should be congratulated for this.

Major concerns.

1. The gene classification in the transcriptome analysis seems to be improper. For example, Idi 1 is not an enzyme involved in Phospholipid synthesis. The list in suppl figure 1 needs to be verified.
2. It is not essential that decrease in lipid metabolism enzymes will result in the decrease of the lipids. This is because both catabolic and anabolic enzymes will be classified as metabolic enzymes. In the pathway shown in fig 1d, there is a pronounced decrease in Ptd Cho catabolic enzyme PLA2 while the changes in it anabolic enzyme is very little. From Supplementary figure 1, the highly downregulated enzymes are phospholipases, indicating that there should be an accumulation of phospholipids. Why is this not happening here?
3. The results showed in figure 1e is not enough to show that PC supplementation improved the myelination. Moreover, Figure 1e and 2g are contradicting each other. There is no change in myelinated axon upon PL treatment while with PC treatment, it increases.
4. The pathway showed in 1d is not clear. Fold changes in TG should be represented there.
5. If the authors used the BODIPY-PC mentioned in the method section, it is not head group labeled but tail labeled. It should be shown that BODIPY-PC supplementation also leads to increased myelination (figure 1f and 2a)
6. Most of the internal standard used in the lipidomics analysis can be present endogenously in myelin. This can affect the quantitation significantly. This needs to be clarified.
7. Figure 3k is confusing. What lipid class were analyzed here? If it is PL how does one get 54? If 54 is from TAGs, having tri-acylated and di-acylated lipids in the same graph did not make sense. This needs to be clarified.
8. The most important issue with this paper is that the reduction in all lipid classes was rescued by PL containing diet without altering the decreased expression of lipid synthesis proteins. The authors suggesting a direct compensation of lipids. However, there are no convincing pieces of evidence to prove this point as;
 - a) Only three lipid metabolizing enzymes were assessed after PL treatment.
 - b) There is no explanation how PL will increase the Cholesterol and Cer content?

9. What is the statistical power in figure 1e? Why there is a difference in the sample size? Similarly one cannot derive statistical significance with two data points as shown in 3e.

Reviewers' comments:

Reviewer #1 (Remarks to the Author):

The study by Fledrich et al. investigates a rat model of CMT1A, with a focus on developing new therapeutic approaches to prevent progressive demyelination and axonal loss in the disease. The study is based on previously published findings in the same model demonstrating that Pmp22 transgenic Schwann cells (SCs) display a significant delay in myelination, with many axons remaining completely unmyelinated until adulthood. These fibers degenerate with time, suggesting that enhancing myelination might rescue axon survival.

In the present study, the authors focus on determining how altered SC differentiation is mechanistically linked to reduced myelination in CMT1A, in particular in the rat model of the disease. The authors demonstrate that abnormalities in SC lipid metabolism is a causative factor in the pathogenesis, as myelin lipids are crucially important for myelin membrane growth and stability/integrity.

In this study, Fledrich et al. use an unbiased approach to demonstrate that SCs in the rat model display a developmental defect that includes reduced transcription of genes required for myelin lipid biosynthesis. As a consequence of this defect, lipid incorporation into myelin is severely reduced, leading to an overall imbalance in the stoichiometry of myelin lipid and proteins, and ultrastructural alterations in myelin itself. The authors also find that supplementation of phosphatidylcholine and phosphatidylethanolamine in the diet is sufficient to rescue myelination deficits of SC in vivo. This treatment not only rescues the cellular and the ultrastructural myelin phenotype, but also improves the clinical phenotype.

Overall, this is a very well executed study with an interesting translational potential. The results also identify a clear biochemical/metabolic defect in SCs of the CMT1A rat model, which is novel. The link between genetic and cellular defects, and the myelin and behavioral phenotype is very interesting and important. I have a few suggestions about expanding the analysis presented in the paper in its current form.

1. It would be important to analyze the effects of diet supplementation with myelin lipids on axons and axonal pathology, including ultrastructural and morphological features – e.g. nodes and paranodes.

We thank the reviewer for pointing to the importance of a detailed description on axonal pathology with respect to the therapeutic effect of lipid supplementation in the CMT1A rat model. We performed additional histological analyses in order to address this point and included these morphological data into the revised version of the manuscript:

In detail, we analyzed the neurofilament density on EM level in CMT1A animals, as an aberrant density of axonal neurofilaments has been previously shown to impair peripheral nerve function and to be a feature not only of CMT subforms with mutations in NF, but also in genes encoding myelin proteins (Yin et al, 1998; PMID: 9482781). Notably, a nearest neighbor analysis revealed

an increased neurofilament density in CMT rats, which was significantly improved after PL diet (new Fig.2I,J). However, the reduced axonal diameter characteristic of CMT1A pathology was unchanged after PL treatment (new Suppl. Fig. 2C). Also, the increased nodal width in CMT1A rats remained unaltered after phospholipid treatment, as assessed in teased fibers with immunohistochemical stainings for Na_v1.6 and the myelin sheath (see new Suppl. Fig. 2G). In addition, we assessed the abnormally reduced internodal length in treated CMT1A rats, which again was not ameliorated upon PL treatment (new Suppl. Fig. 2F).

We conclude that, next to improved myelination, PL treatment improves the axonal neurofilament architecture, which may contribute to an improved nerve function in treated CMT1A rats (see page 17 line 27 to page 18 line 14).

2. Early and late treatment with myelin lipids. These are not only different in timing but also in duration. The early treatment is from P2 to P21, and the late treatment is from P21 to P90. It would be interesting to determine whether a short-duration late treatment would also be effective. Is the long duration necessary?

We agree with the reviewer that knowledge about the dynamics of a late treatment is important for a putative translation of a therapeutic strategy into the clinics. We therefore added a longitudinal phenotype analysis of CMT rats that were treated with PL from P21-P90. In the new Fig. 4J we demonstrate that the treatment effect of PL therapy is first visible at P90, but not yet at P35 and P63, indicating that long term treatment in adulthood is required to exert a therapeutic effect (see page 21, line 14-15 of the revised manuscript).

Reviewer #2 (Remarks to the Author):

The authors use a transgenic rat overexpressing peripheral myelin protein 22 (Pmp22), which develops Charcot-Marie-Tooth 1A disease, in order to characterize and potentially prevent its neuropathy phenotype. They use a combination of transcriptomic and lipidomic analysis to demonstrate altered lipid metabolism in peripheral nerves of affected animals. Based on these data they design and evaluate the efficiency of phospholipid supplementation therapy. Interestingly, they demonstrate that both long-term and relatively short-term dietary supplementation of phosphatidylcholine and phosphatidylethanolamine partially alleviate the neuropathy phenotype in transgenic animals. The presented data are very interesting with regard to the potential of this approach to be translated into clinical trials. However, the underlying mechanistic part of the story is somehow less clear.

1. The transcriptomic data presented in Figure 1 in part overlap with the data previously published by the same group (Figure 2, Fledrich et al., Brain 2012). The differences and/or similarities between the results from the two studies should be discussed. Also, the previous characterization of CMT1A rat model by the authors (Fledrich et al., Brain 2012) revealed a high variability in the severity of the disease, similar to patients with CMT1A. Did the authors consider to stratify affected animals (mildly vs severely affected) in order to potentially decrease variability in the observed phenotypes pre/post treatment?

We thank the reviewer for pointing to the need of clarification of similarities/differences to our previous transcriptomic analyses using the CMT rat model. In our 2012 paper (Fledrich et al., Brain 2012) we performed comparative transcriptomics (using microarrays) between mildly and severely affected CMT rats at one single early (P6) and one single late (P90) timepoint. We indeed already found lipid associated genes to be differentially expressed between mildly and severely affected CMT rats, which allowed us to derive surrogate biomarkers for disease severity (Fledrich et al., 2012). In the present study, however, we performed longitudinal RNAseq analyses between wildtype and CMT rats during the course of myelination. This experimental design allowed us to extract a hypothesis about the early disease development during myelination. For clarification, we added a respective paragraph to the revised manuscript (see page 3, line 22-26 and page 14, line 28-30).

With respect to the second part of the reviewers question on the experimental design for the treatment trials we added a more precise description into the method section of the revised manuscript. In detail, we performed a-priori power analysis integrating the disease variability in order to calculate the required group sizes. In addition we have stratified all CMT rats for the late longterm treatment (P21-P90) before therapy start (P19). However, for the trials starting right after birth (P2-P21 and P2-P90), stratification was not possible (as phenotype tests at that age are not applicable), which hence explains the observed variability. In these trials, rats were randomly allocated to the different treatment groups. We added a respective statement to the revised manuscript (see page 5, line 17-22).

2. The observed effect of phosphatidylcholine (PC) in myelinating cultures (Figure 1b-e) is very interesting and could be explored a bit more. Would the same treatment also work in standard media (not media with delipidated serum)?

In the standard protocols for myelinating cocultures fetal calf serum is used, which contains an extremely high concentration of different lipids. While this ensures a saturation of the media with exogenous lipids which indeed is useful to obtain comparable cultures with respect to different scientific questions, the artificial oversupply of Schwann cells with lipids prevents the investigation of lipid-metabolism related research questions in vitro (Saher et al., 2009, PMID: 19439587; Giles et al., 1981, PMID: 7249360).

In detail, standard culture media with 10% fetal calf serum (Hyclone, ThermoFisher) contains ~10µg/ml cholesterol, 11µg/ml phosphatidylcholine and more than 70µg/ml neutral lipids thereby masking any effect of a specific lipid supplementation (in our case 2µg/ml PC). In line, when we repeated the coculture experiments with standard media (with 10% FCS) in response to the reviewers question, we found no treatment effect on myelination in vitro, neither in wildtype nor in transgenic cocultures (**new Suppl. Fig. 1D, E**). We added a respective statement to the revised manuscript (page 15 line 30 to page 16 line 1).

3. What happens with myelin protein and cholesterol biosynthesis (both presumably needed for myelination) in PC-treated cultures?

We thank the reviewer for pointing to the important aspect of myelin biosynthesis upon PC treatment, which indeed also requires myelin protein and cholesterol.

*Within the original version of the manuscript, we have demonstrated that PC treatment resulted in an improved myelination without activation of the abnormally decreased PI3K/AKT signaling pathway and without a regain in lipid mRNA expression (old **Fig. 4F-G**). We hence proposed that the therapeutic effect is mediated by a direct compensation of the endogenous failure of Schwann cells to mount lipid metabolism. In order to further verify this point, we performed a new, large scale in vivo qPCR analysis after PL therapy which indeed confirmed that PL treatment does not induce a transcriptional rescue of myelin and lipid genes (**new Fig. 4G**). In line, we observed no transcriptional upregulation of the rate limiting enzymes for cholesterol biosynthesis (*Hmgcr* and *Scap*), as well as for the myelin protein *Mpz* in vitro after PC treatment of myelinating co-cultures (**new Suppl. Fig. 3F**). Only, a slight upregulation in wildtype cocultures was found for *Mbp* mRNA in vitro (**new Suppl. Fig. 3F**).*

*We hence conclude that the promyelinating effects of phospholipids in vivo and in vitro are most likely downstream and independent of a transcriptional regulation and that Schwann cells execute cholesterol and myelin protein biosynthesis via post-transcriptional changes by compensation and regulation on the protein level (e.g. enzymatic activities). In order to demonstrate in proof-of-principle that PC treatment indeed results in an increased biosynthesis of the other myelin components, we took advantage of differentiating primary rat Schwann cell monocultures to measure the cholesterol production as a function of PC treatment. Importantly, we found more cholesterol to be produced in Schwann cells treated with PC (**new Fig. 4H-I**), without a concomitant change in cholesterol gene expression (**new Fig. 4H-I**).*

We added a respective paragraph to this point together with the new data sets to the revised version of the manuscript (see page 20 line 25 to page 21 line 5 and page 22 line 25-30).

4. The electrophysiological data presented in Figure 2 are difficult to interpret. Nerve conduction velocity (NCV) is only slightly increased by phospholipid therapy in CMT1A rats. However, the number of myelinated axons (which is the contributing factor for NCV) is almost completely normalized. Increase in compound muscle action potential after phospholipid therapy may also indicate changes in neuromuscular junctions in addition to observed changes in muscle – this aspect could be evaluated.

Our electrophysiological measurements revealed an increase in compound muscle action potential (CMAP) as well as in nerve conduction velocity (NCV) in CMT rats after PL therapy. Of note, the NCV is in general predominantly determined by the parameters myelination, fiber diameter and internodal length and does not correlate well with numerical changes in the amount of myelinated axons, as only the fastest population of axons and not all axons are captured by standard NCV measurements (see Mallik and Weir, 2005, PMID: 15961865; Li 2015, PMID: 25792482). In contrast, the CMAP predominantly reflects the number of functional axons (and hence, neuromuscular transmissions) per nerve (see Mallik and Weir, 2005, PMID: 15961865; Li 2015, PMID: 25792482). Hence, our findings are well in accordance with the literature as the rescue of the number of myelinated fibers per nerve after PL treatment in CMT

*rats results in an increased CMAP. The improved NCV, however, may be due to the improved molecular composition (and hence, insulation) of the myelin sheath, as fiber diameter (**new Suppl. Fig. 2C**) and internodal length (**new Suppl. Fig. 2F**) remained unaltered in response to PL treatment. We added the respective citations and the new data to the revised version of the manuscript (page 17, line 23; page 18 line 3-15).*

5. As described by the authors, the affected animals present a mix of hypomyelinating, amyelinating and hypermyelinating axons (Figure 3). Also, it was previously shown that there is a more pronounced defect in ventral roots (Sereda et al., Neuron 1996). The characterization of morphological changes in Figure 3a-c could therefore be more informative if done separately on roots (in particular ventral roots, containing motor fibers). Also, a more detailed quantification of different abnormalities could be done (e.g. number of unmyelinated axons, number of onion bulbs) since the heterogeneity of the various abnormalities in myelinating Schwann cells may obscure presented g-ratio data.

*As requested by the reviewer, we performed a more detailed electron microscopical analysis of peripheral nerves in CMT rats after PL therapy. Whereas we did unfortunately not collect spinal nerves including ventral roots at study end, we now included morphological data on tibial nerves with regard to number of unmyelinated axons (**new Suppl. Fig. 2D**), axonal size (**new Suppl. Fig. 2C**) and ultrastructure (**new Fig. 2I,J**) as well as a quantification of onion bulbs (**new Suppl. Fig. 2E**). Here, we found an (1) improved neurofilament density (which is aberrantly increased in CMT1A) after PL therapy without changes of the axonal diameter. (2) We did not detect an effect of PL treatment on unmyelinated axons and onion bulb formations, in line with our hypothesis that PL therapy functions by a direct compensation of the endogenous failure of myelinating Schwann cells to mount lipid metabolism. We added a respective description into the revised manuscript (see page 17 line 27 to page 18 line 14).*

6. How does the observed improvement in myelin interperiodic distance in animals treated with phospholipids (Figure 3d-g) connect to concomitant electrophysiological and behavioral changes?

*We thank the reviewer for pointing to clarification of the relation of myelin ultrastructural changes and the observed improvements in electrophysiology and motor phenotype. Whereas the higher number of myelinated fibers per peripheral nerve after PL treatment in CMT rats unequivocally contributes to the improved clinical and electrophysiological phenotype, we agree with the reviewer that the consequences of improved myelin compaction in CMT rats are difficult to decipher. It may be plausible, that an optimized molecular composition and architecture of myelin would reduce capacitance and increase resistance in nerve fibers and, hence, improve nerve conduction (Snaidero and Simons, 2014, PMID: 25024457). Since we could not observe any changes after PL therapy for the main parameters that influence nerve conduction - myelin sheath thickness (**Fig. 3A-B**), myelin internodal length (**new Suppl. Fig. 2F**) and axon caliber (**new Suppl. Fig. 2C**) - we assume that the improved molecular composition and ultrastructure could indeed underlie improved nerve conduction. We added a more extensive discussion of this point to the revised version of the manuscript (page 23 line 31 to page 24 line 4).*

7. There is an increase in the number of myelinated axons in CMT1A animals on phospholipid therapy (Figures 2g and 4c). From where are these fibers coming?

In a previous study we examined the development of myelination and the number of myelinated fibers in CMT rats (Fledrich et al., 2014, PMID: 25150498). We found that CMT1A rats never achieve a normal number of myelinated axons during postnatal development. Those fibers that should be myelinated (caliber >1 μ m) but remain amyelinated during development survive within the nerve until later disease stages in adulthood, and only slowly degenerate with disease progression (Fledrich et al., 2014). We therefore believe that PL therapy helps Schwann cells to physiologically myelinate this pool of amyelinated fibers >1 μ m and thereby protects these fibers against degeneration. We discussed this point in the revised manuscript (page 23, line 6-14).

8. Also, there is no improvement in decreased transcription of lipid genes in CMT1A animals (Figure 4g). While the phospholipid therapy may provide sufficient amounts of phosphatidylcholine and phosphatidylethanolamine for additional myelin, it is unclear from where cholesterol and myelin proteins are coming, which are the main components of the myelin membrane.

We thank the reviewer for pointing to the very important aspect of how Schwann cells can produce other myelin compounds besides the supplemented phospholipids and also refer to our answer to question #3.

*In detail, we have strongly extended the qPCR dataset after PL treatment with regard to genes involved in lipid catabolism and anabolism (**new Fig. 4G**). Importantly, although we confirmed an overall downregulation of those genes in peripheral nerves of CMT rats, we could not detect any alteration of lipid gene expression after PL therapy. We conclude that the observed improved myelination is supported by PL therapy most likely independent of gene expression regulation, which is also in line with a persistent decrease of PI3K/AKT signaling after therapy, a pathway crucial for Schwann cell lipid gene transcription (Domènech-Estévez, et al., 2016, PMID: 27098694; Taveggia, 2016, PMID: 27089429). An increase in the production of other myelin compounds besides phospholipids therefore requires post-transcriptional mechanisms, e.g. on the protein level of enzymatic activity. To collect insight into such possible mechanisms, we performed a proof of principle experiment and treated differentiating primary rat Schwann cell monocultures with phosphatidylcholine and measured cholesterol and cholesterol gene expression after 2 days in vitro. Indeed, we robustly found more cholesterol in Schwann cells that were treated with PC (**new Fig. 4H-I**), without a concomitant change in cholesterol gene expression (**new Fig. 4H-I**). We conclude that the therapeutic effect of PL treatment of CMT rats acts downstream of lipid gene expression and that Schwann cells are indeed able to execute cholesterol biosynthesis in response to phospholipid treatment in CMT1A (see page 20 line 25 to page 21 line 5 and page 22 line 25-30 of the revised manuscript).*

Minor comments:

In Figure 1f, please present the images with the same magnification as in 1b in order to have a more global view of myelin segments.

*We have included new pictures in **Fig.1f** with a more global view and also included boxes with higher magnification.*

The co-labeling between MBP and BODIPY-PC (Figure 2a) could be done on teased fibers or on nerve cross sections in order to improve resolution.

*In order to improve resolution for colocalization of MBP and BODIPY-PC, we repeated the imaging with high resolution confocal microscopy. A **new Fig. 2a** is included into the manuscript.*

Reviewer #3 (Remarks to the Author):

The paper by Fledrich et al. describes changes in lipid composition and synthesis in Charcot-Marie-Tooth type 1A, caused by increased gene dosage of PMP22. The paper demonstrates that these changes are the down stream consequence of changes in transcription levels of the lipid biosynthetic enzymes. The paper goes on to demonstrate that exogenously applied lipids can improve myelination in vitro, and can incorporate into myelin sheaths in vivo after IV injection. The therapeutic potential of this approach is examined in vivo, and modest but significant improvements in myelin and motor performance are described. The time course of these improvements is also examined, and it is determined that the lipid dosing must be maintained to maintain the improvement, but that some improvement can be found even with dosing beginning after development. Most of these experiments were done using a very good rat transgenic model of CMT1A.

The paper is well written and the results are interesting and supported by the data presented. The rationale for the in vivo experiments is well established by the transcriptome and in vitro studies. The in vivo studies use rigorous study design, with randomization of animals into treatment groups, pre-established inclusion and exclusion criteria, and single blind collection of the data. The results are generally well presented (but see points below) and the outcomes are positive, but generally not overstated. The assessment of CMAP amplitudes and myelinated axon numbers are the most convincing data. The straightforward nature of the proposed intervention is appealing.

Addressing the comments below would further improve the manuscript.

1. The g-ratio results shown in figure 3C are problematic as presented. The relationships between axon size and myelin thickness shown in Figure 3B are the more appropriate indicators of what is happening, and also indicate that PL treatment did not change this relationship. The

average g-ratio shown in 3C is just an indirect indicator that there are more small (hypermyelinated) axons than large (hypomyelinated) axons. Panel 3C should simply be removed. In Figure 3F, a picture of the myelin packing of a PL treated nerve would improve the figure.

*We thank the reviewer for this constructive suggestions and we have removed **Fig. 3C** and included a representative picture for the treatment group in **new Fig. 3F**, accordingly.*

2. The age at which myelinated axon number first becomes significantly lower than control is an important point and should be reported. This matters for interpreting the improvements in myelinated axon number with treatment starting at P21 (Figure 4L).

We agree with the reviewer that the information about the disease course and at which age the difference in the number of myelinated fibers first becomes obvious is very important. We previously have thoroughly analyzed the time course of myelination in the CMT rat (Fledrich et al., 2014). We found that the reduced number of myelinated axons can first be observed at P18. However, fibers are not lost at this time point yet and can still be myelinated from P21 on. We have included this information into the manuscript (page 23, line 6-14).

3. The improvements in muscle size are problematic. This is particularly true in Figure 4I, where wild type rats treated with PL are not shown, but also in 2D, where the magnitude of the effect of PL on wild type muscles seems comparable to the improvement seen in mutant muscles, but without the statistical significance. The wild type with 3% PL treatment appears to be the result with the lowest number of animals in 2D.

*For the late treatment paradigm from P21-P112 in **Fig. 4h-i**, we did not include a wildtype PL treatment group, as we did not detect treatment effects in phenotype, electrophysiology and histology in treated wildtype rats in the P2-P112 paradigm (**Fig. 2c, e, f, g**). We agree however with the reviewer that the mean value in muscle circumference may appear somewhat increased in wildtypes treated with 3% PL. However, the individual animals showed a high variation of the muscle circumference, which indeed explains the difference in the statistical significance compared to the treated *Pmp22* tg animals. In order to further evaluate a potential effect of PL treatment on wildtype muscle circumference we again treated wildtype rats with 3%PL from P2-P112 aiming at increasing the n-number and to strengthen the statistical analysis of the wildtype data. Of note, the inclusion of new treated (n=10) versus non-treated (n=10) animals confirms that 3% PL treatment does not significantly (p=0.99) increase the muscle circumference in wildtype animals. We have used this data to fill up the groups in the **new Figure 2d**.*

4. Are phospholipids passed in the milk of nursing dams? The authors acknowledge that rat pups do not eat solid food until P15, which is important for interpreting the results, especially in figure 4, but whether ingestion by the mother is effectively passed to the pups should be noted.

We agree with the reviewer that information about the supplemented lipids passed in the milk of the nursing dams is important. It is well established that breast milk composition is extremely

sensitive to maternal nutrition (Innis, 2014, PMID: 24500153 ; Oosting et al., 2015, PMID: 26097702), and that a high fat diet leads to the production of high fat milk (Nicholas and Hartmann, 1991, PMID: 1674460). However, to investigate to which extent the 3%PL enriched diet has modified the milk composition, we have treated the nursing dams after they gave birth from P2-P9. At P9, we collected the milk from the stomachs of the pups and subjected the samples to mass spectrometry. The lipid proportion of milk in general consists out of >98% neutral lipids, mostly TAG (Jensen et al., 1991, PMID: 1779072). In our study, with over 60%, the major fatty acid compound in the supplemented phospholipid (SIGMA, #P3644) is linoleic acid, which harbors two double bonds (C18:2). When plotting the analyzed milk neutral lipids (TAG plus DAG) as a function of the double bonds, a significant shift towards species with either two or four double bonds is visible, which clearly indicates that the supplemented phospholipid fatty acid tails have passed the mother's milk in the form of neutral lipids (new Suppl. Fig. 2B, see page 17, line 1-11).

5. The Discussion is well written and generally ties the results into the larger context well, especially around previous work in the mTOR/AKT pathway. However, the authors should also include a brief discussion of their findings in comparison to the recent JCI paper by Zhao et al., showing that downregulation of PMP22 transcript levels using a gene therapy approach is also beneficial. Key points seem to be the post-onset benefits seen by Zhao in comparison to those reported here, and the possibility of combining these approaches.

We agree with the reviewer, that the study by Zhao et al. constitutes an important aspect to be discussed in our manuscript and we included a respective paragraph (page 24, line 18-22).

Minor points:

A reference for the PMP22 transgenic mouse strain used for the Schwann cell cultures should be provided in the methods section.

We thank the reviewer for making us aware that we only have referenced the transgenic mouse strain in the cell culture part of the methods section. It is now also referenced in the animal part.

Reviewer #4 (Remarks to the Author):

The manuscript titled "Targeting myelin lipid metabolism as an effective therapeutic strategy in a rodent model of CMT1A neuropathy" by Fledrich et. al describes the role of lipid metabolism in Charcot-Marie Tooth Disease 1A. Using "omics" approach the authors showed a decrease in lipid metabolism gene expression as well as lipid content in myelin of pmp22 transgenic rats. Besides providing a plausible missing link in CMT1A pathogenesis, this results offer a therapeutic target in treating CMT1A. Using an easily translatable therapeutic approach, phospholipid supplementation, the authors showed improvement of CMT1A. This manuscript provides yet another evidence to the importance of lipids in the nervous system. Also, this

manuscript showed how an omics data can be translated into a potent therapeutic application. The authors should be congratulated for this.

Major concerns.

1. The gene classification in the transcriptome analysis seems to be improper. For example, Idi 1 is not an enzyme involved in Phospholipid synthesis. The list in suppl figure 1 needs to be verified.

The gene ontology (GO) gene sets were obtained from the Molecular Signatures Database (MSigDB) from the Broad Institute UC San Diego (<http://software.broadinstitute.org/gsea/msigdb/index.jsp>), a well curated and annotated database that is widely accepted and used for transcriptome analyses. The gene sets were extracted from MSigDB without any further manipulation and the identifiers are given in the figure legend. We discussed this point with bioinformatical experts in response to the reviewers question and strongly believe that it is incorrect to manipulate given gene sets. We would hence prefer to abstain from any modification of the original gene sets. In order to clarify this aspect within the manuscript, we added a more precise description of the origin of the gene sets into the revised manuscript (see page 13, line 26-30).

2. It is not essential that decrease in lipid metabolism enzymes will result in the decrease of the lipids. This is because both catabolic and anabolic enzymes will be classified as metabolic enzymes. In the pathway shown in fig 1d, there is a pronounced decrease in Ptd Cho catabolic enzyme PLA2 while the changes in it anabolic enzyme is very little. From Supplementary figure 1, the highly downregulated enzymes are phospholipases, indicating that there should be an accumulation of phospholipids. Why is this not happening here?

To our opinion, RNAseq is powerful to understand global changes in gene expression. We could demonstrate that genes related to lipid biosynthetic process display an overall downregulation in CMT rat peripheral nerves (despite single genes may be up) and we would feel confident to conclude reduced lipid biosynthesis is a feature of CMT1A disease in transgenic rats. However, the reviewer's question of why no PC is accumulating when the expression of the catabolizing PLA2 is reduced may be answered by the likewise reduced anabolizing enzymes. We added a respective note to the revised version of the manuscript (page 15, line 5-6). Of note, the RNAseq data are also in agreement with the lipidomics approach, where we could show that indeed less lipids are incorporated into the myelin sheath.

3. The results showed in figure 1e is not enough to show that PC supplementation improved the myelination. Moreover, Figure 1e and 2g are contradicting each other. There is no change in myelinated axon upon PL treatment while with PC treatment, it increases.

We thank the reviewer for pointing to the variation in response to PC supplementation in vitro in figure 1c, which is the result from single cultures derived from individual E13.5 embryos from two timed pregnant females (18 embryos in total) in parallel. We choose this experimental

*design to avoid slight differences in handling of subsequently grown cultures. We randomly distributed wt and tg embryos to either the treatment or the control group (there were 11 WT and 7 TG embryos). By that time, only one cover slip per embryo was produced, which, however, apparently displays certain overall variation within the culture system. In order to unequivocally demonstrate that PC treatment indeed increases myelination in vitro (in addition to the increased myelination by PL in vivo) we now performed additional cell culture experiments with two paired cover slips from each Pmp22 tg Embryo (n=7), of which one was treated with PC, whereas the other one served as control (**new Suppl. Fig 1E**). Like before, also this approach revealed a significant increase of myelinated segment in response to PC treatment.*

With respect to the second point of the reviewer's question, we note that Fig.1e and 2g display results from in vitro and in vivo experiments, respectively. Whereas the results within the Pmp22tg groups between both experiments are coherent, we agree with the reviewer that the wildtype rats in vivo do not show increased myelination in response to PL as they do in response to PC in vitro. We note that in vitro, the cultures are grown under starvation in lipoprotein deficient media, whereas this is not the case in vivo. Therefore, cultured wildtype and Pmp22tg Schwann cells could assimilate PC, incorporate it into myelin and use it to improve myelination. On the other hand, the lipid biosynthesis machinery in vivo in the wildtype situation is likely to already tap the full potential. We added a respective discussion of this point to the revised version of the manuscript (page 22 line 33 to page 23 line 5).

4. The pathway showed in 1d is not clear. Fold changes in TG should be represented there.

*We have added the fold change values to **Fig. 1d** accordingly.*

5. If the authors used the BODIPY-PC mentioned in the method section, it is not head group labeled but tail labeled. It should be shown that BODIPY-PC supplementation also leads to increased myelination (figure 1f and 2a)

We thank the reviewer for making us aware about the mistake in the manuscript about the labelled PC, which is indeed tail-labelled BODIPY-PC. We corrected the manuscript accordingly.

*The purpose of the experiments using BODIPY-PC in vitro (Fig. 1f) and in vivo (Fig. 2a) was to demonstrate that PC can be taken up and utilized by Schwann cells to be integrated into the myelin sheath. That PC increases myelination in CMT1A is shown thoroughly in vitro in **Fig.1e** and **new Suppl. Fig. 1E**, and for PL extensively in vivo in **Fig.2 - Fig.4**.*

6. Most of the internal standard used in the lipidomics analysis can be present endogenously in myelin. This can affect the quantitation significantly. This needs to be clarified.

The reviewer addresses an important issue in MS-based lipid quantification using internal lipid standards. To test for a potential overlap of endogenous lipids and standards we have

performed lipid extractions in the absence of lipid standards and subjected the samples to MS analysis and data evaluation. Our results show that the lipid standards used for MS analysis do not significantly overlap with endogenous lipid species. As an example, the major phosphoglycerolipids species phosphatidylcholine and phosphatidylethanolamine showed an overlap of standards with endogenous species of 0.1% and 0.5%, respectively. The results are summarized in the **new Suppl. Table 1**. In average, signal intensities of $0.82 \pm 0.7\%$ were found to be present in endogenous lipid samples that overlapped with the m/z values of standards. We also added a respective paragraph into the revised manuscript (page 11, line 24 to page 12 line 11).

7. Figure 3k is confusing. What lipid class were analyzed here? If it is PL how does one get 54? If 54 is from TAGs, having tri-acylated and di-acylated lipids in the same graph did not make sense. This needs to be clarified.

We thank the reviewer for pointing out that this figure needs more clarification. In Fig 3l, all lipid molecules (from all classes in Fig. 3k, including TAG, DAG, PC, PE, etc.) per sample were analyzed for their acyl carbon number. However, to reduce the risk of confusion, in the **new Fig. 3J** we do only show lipid molecules with 36 acyl carbons (most likely glycerolipids with two C18 acyl chains), which are less abundant in CMT rats, but normalized after PL treatment. As the experimental PL diet was especially enriched in C36 phospholipids, we conclude that the dietary lipids have reached the myelin compartment.

8. The most important issue with this paper is that the reduction in all lipid classes was rescued by PL containing diet without altering the decreased expression of lipid synthesis proteins. The authors suggesting a direct compensation of lipids. However, there are no convincing pieces of evidence to prove this point as; a) Only three lipid metabolizing enzymes were assessed after PL treatment. b) There is no explanation how PL will increase the Cholesterol and Cer content?

Within the original version of the manuscript, we have demonstrated that PC treatment resulted in an improved myelination without activation of the abnormally decreased PI3K/AKT signaling pathway and without a regain in lipid mRNA expression (old Fig. 4F-G). We hence proposed that the therapeutic effect is mediated by a direct compensation of the endogenous failure of Schwann cells to mount lipid metabolism.

In order to further verify this point, we have strongly extended the qPCR dataset after PL treatment with regard to genes involved in lipid catabolism, anabolism and lipid class interconversion (**new Fig. 4G**). Importantly, although we confirmed an overall downregulation of those genes in peripheral nerve of CMT rats, we could not detect any alteration of lipid gene expression after PL therapy.

In line, we observed no transcriptional upregulation of the rate limiting enzymes for cholesterol biosynthesis (*Hmgcr* and *Scap*), as well as for the myelin protein *Mpz* in vitro after PC treatment of myelinating co-cultures (**new Suppl. Fig. 3F**). Only, a slight upregulation in wildtype cocultures was found for *Mbp* mRNA in vitro (**new Suppl. Fig. 3F**). We hence conclude that the

promyelinating effects of phospholipids in vivo and in vitro are downstream of a transcriptional regulation and that Schwann cells execute cholesterol and myelin protein biosynthesis via post-transcriptional changes by compensation and regulation on the protein level (e.g. enzyme activities). To collect insight into such possible mechanisms, we performed a proof of principle experiment and treated differentiating primary rat Schwann cell monocultures with phosphatidylcholine and measured cholesterol and cholesterol gene mRNA expression after 2 days in vitro. Indeed, we robustly found more cholesterol in Schwann cells that were treated with PC (new Fig. 4H-I), again without concomitant upregulation of cholesterol genes (new Fig. 4H-I).

We added a respective discussion of this point together with the new data sets to the revised version of the manuscript (see page 20 line 25 to page 21 line 5 and page 22 line 25-30).

9. What is the statistical power in figure 1e (1)? Why there is a difference in the sample size (2)? Similarly one cannot derive statistical significance with two data points as shown in 3e (3).

*(1) The statistical power was computed by G*Power 3.1.9.2. The settings were: one-way ANOVA, fixed effects, effect size from means; alpha error: 0.05; total sample size 18; 4 groups (mean values and group sizes were entered) ; average SD: 60.8; calculated effect size: 1.2422; eta-square: 0.584; power: 0.9792.*

(2) The difference in sample size between the groups derive from the nature of the experiment. Our experience has shown that quantification of cocultures works best if the cultures were grown in parallel (not subsequently). To avoid variation that would derive from slightly different handling of subsequently grown cultures, the experiment was designed to process the E13.5 embryos from two timed pregnant females (18 embryos in total) in parallel. Cultures from WT and TG embryos were randomly distributed to either the treatment or the control group (there were 11 WT and 7 TG embryos) and by that time only one cover slip per embryo was produced. However, we have now performed additional cell culture experiments with two paired cover slips from each Pmp22tg Embryo (n=7), of which one was treated with PC, whereas the other one served as control (new Suppl. Fig 1E). Like before, also this approach revealed a significant increase of myelinated segment in response to PC treatment.

(3) We thank the reviewer for this point and have removed the statistical test from the box plot in figure 3e accordingly.

REVIEWERS' COMMENTS:

Reviewer #1 (Remarks to the Author):

The authors have made a significant effort to address all the issues raised in the previous review cycle. The paper has been extensively revised and significant amount of new data has been added. Many points that required clarification and extensive re-writing have also been addressed. I still believe that the overall quality of the work is very high, and that this is a novel and significant contribution to the field. Therefore, I believe that the paper should now be acceptable for publication in Nature Communications.

Reviewer #2 (Remarks to the Author):

In the revised version of their manuscript the authors provide substantial additional data that are addressing the majority of my comments.

I have only a few minor remarks:

- 1) Legend of Figure 1f: the authors should not refer to "upper/lower panels" since the figure was rearranged.
- 2) Legends of Figures 2g and h seem to be inverted.
- 3) Legend for Figure 3h: the text should probably read "Quantification of g shows...."
- 4) Supplementary Figure 2d: the authors may like to synchronize the terminology between "amyelinated axons" used in the figure panel and "unmyelinated axons" used in the figure legend.

Reviewer #3 (Remarks to the Author):

My previous concerns with the paper by Fledrich et al. have been addressed in this revised manuscript. The authors use rigorous preclinical study design to examine the effects of phospholipid supplementation in a rat model of CMT1A. The outcomes are interesting and are not over stated. The mechanism is plausible based on prior work and data presented here, and the authors are largely able to account for the treatment effects they see with corresponding biochemical and anatomical changes. The proposed translational potential merits further investigation given the rather straightforward, low cost and low risk nature of dietary supplementation with lipids. The technical points raised in the first submission by all reviewers have been adequately addressed. There are a few minor language inconsistencies that do not detract from the ability of the reader to understand the paper, but that should be addressed in copy editing.

Reviewer #4 (Remarks to the Author):

The authors responded to most of my comments convincingly. Now the results support the author's claims clearly. However, it is hard to understand the supplementary table. 1 (mass spectrometry internal standard) without any associated description or legends. The table description should be added.

REVIEWERS' COMMENTS:

Reviewer #1 (Remarks to the Author):

The authors have made a significant effort to address all the issues raised in the previous review cycle. The paper has been extensively revised and significant amount of new data has been added. Many points that required clarification and extensive re-writing have also been addressed. I still believe that the overall quality of the work is very high, and that this is a novel and significant contribution to the field. Therefore, I believe that the paper should now be acceptable for publication in Nature Communications.

Reviewer #2 (Remarks to the Author):

In the revised version of their manuscript the authors provide substantial additional data that are addressing the majority of my comments.

I have only a few minor remarks:

- 1) Legend of Figure 1f: the authors should not refer to “upper/lower panels” since the figure was rearranged. *Corrected.*
- 2) Legends of Figures 2g and h seem to be inverted. *Corrected.*
- 3) Legend for Figure 3h: the text should probably read “Quantification of g shows....” *Corrected.*
- 4) Supplementary Figure 2d: the authors may like to synchronize the terminology between “amyelinated axons” used in the figure panel and “unmyelinated axons” used in the figure legend. *Corrected.*

Reviewer #3 (Remarks to the Author):

My previous concerns with the paper by Fledrich et al. have been addressed in this revised manuscript. The authors use rigorous preclinical study design to examine the effects of phospholipid supplementation in a rat model of CMT1A. The outcomes are interesting and

are not over stated. The mechanism is plausible based on prior work and data presented here, and the authors are largely able to account for the treatment effects they see with corresponding biochemical and anatomical changes. The proposed translational potential merits further investigation given the rather straightforward, low cost and low risk nature of dietary supplementation with lipids. The technical points raised in the first submission by all reviewers have been adequately addressed. There are a few minor language inconsistencies that do not detract from the ability of the reader to understand the paper, but that should be addressed in copy editing.

Reviewer #4 (Remarks to the Author):

The authors responded to most of my comments convincingly. Now the results support the author's claims clearly. However, it is hard to understand the supplementary table 1 (mass spectrometry internal standard) without any associated description or legends. The table description should be added.

We added a respective caption to supplementary table 1, accordingly.